# Evaluation of Hand-Crafted Feature Extraction for Fault Diagnosis in Rotating Machinery: A Survey

**DOI:** 10.3390/s24165400

**Published:** 2024-08-21

**Authors:** René-Vinicio Sánchez, Jean Carlo Macancela, Luis-Renato Ortega, Diego Cabrera, Fausto Pedro García Márquez, Mariela Cerrada

**Affiliations:** 1GIDTEC, Universidad Politécnica Salesiana, Cuenca 010105, Ecuador; jmacancelap@est.ups.edu.ec (J.C.M.); mcerrada@ups.edu.ec (M.C.); 2School of Mechanical Engineering, Dongguan University of Technology, Dongguan 523000, China; diego@dgut.edu.cn; 3Ingenium Research Group, Universidad Castilla-La Mancha, 13071 Ciudad Real, Spain; faustopedro.garcia@uclm.es

**Keywords:** condition monitoring indicator, fault diagnosis, frequency domain, gears and bearings, hand-crafted features survey, signal processing, time domain

## Abstract

This article presents a comprehensive collection of formulas and calculations for hand-crafted feature extraction of condition monitoring signals. The documented features include 123 for the time domain and 46 for the frequency domain. Furthermore, a machine learning-based methodology is presented to evaluate the performance of features in fault classification tasks using seven data sets of different rotating machines. The evaluation methodology involves using seven ranking methods to select the best ten hand-crafted features per method for each database, to be subsequently evaluated by three types of classifiers. This process is applied exhaustively by evaluation groups, combining our databases with an external benchmark. A summary table of the performance results of the classifiers is also presented, including the percentage of classification and the number of features required to achieve that value. Through graphic resources, it has been possible to show the prevalence of certain features over others, how they are associated with the database, and the order of importance assigned by the ranking methods. In the same way, finding which features have the highest appearance percentages for each database in all experiments has been possible. The results suggest that hand-crafted feature extraction is an effective technique with low computational cost and high interpretability for fault identification and diagnosis.

## 1. Introduction

Industrial machinery, such as gearboxes, transmission shafts, or reciprocating compressors, are essential rotating equipment widely used in different industries due to their ability to generate force and movement, making them the heart of any mechanical system [1,2]. A machine designed to perform some specific function is expected to do so throughout its useful life. However, a machine may fail due to circumstances often outside our control. We can highlight mechanical parts such as gears, bearings, shafts, belts, or valves as standard components that may be susceptible to failure [2,3,4,5]. Failure or cessation of machine operation can represent significant monetary losses and affect the safety of plant personnel; therefore, the machine must be maintained to prevent such failures [6]. Three primary maintenance schemes are followed worldwide: Reactive, Preventive, and more recently, Predictive [7]. Preventive maintenance has traditionally been the most common maintenance policy in industries. Utilizing this strategy, components are replaced once a specific use time has elapsed. Predictive maintenance prevents failures by constantly monitoring the state of the system and identifying abnormal conditions of machine parts [8]. Predictive maintenance can be divided into Reliability-Centered Maintenance (RCM) and Condition-Based Maintenance (CBM) [9]. CBM is a maintenance approach that uses advanced technologies to continuously monitor and assess the condition of the machinery. In addition, CBM plans necessary maintenance based on the information provided [10]. Due to the need for constant monitoring, the CBM uses a data-based paradigm of three stages: data acquisition, data processing, and diagnosis and decision-making [11,12]. CBM relies on robust and reliable fault diagnosis capabilities [13]. The main tasks of fault diagnosis as a final stage of CBM are indicator selection, identification and determining the cause of a problem or machine fault, intervention planning, monitoring, and evaluation [14,15].

The selection of suitable indicators is crucial for all these tasks to ensure that relevant problems and failures in machinery or equipment are detected. These condition indicators are usually physical or chemical measurements of the equipment used to assess machine conditions and predict the probability of failure. Some typical condition variables are vibration, temperature, and electric current, among others, in the form of temporal series or signals [16,17,18,19,20,21]. Fault diagnosis focuses on analyzing those signals of the machinery state. Changes in measured signals can indicate a problem or failure of the machinery. Constant monitoring of the machine’s condition will support diagnostic processes, associating component failures with processed information. Once the signals for condition monitoring have been selected, the next step to reach a fault diagnosis is processing and analyzing the collected data. Different techniques can be used to analyze condition monitoring signals, such as statistical analysis, time series analysis, and artificial intelligence-based analysis, among others [22,23,24,25,26]. Each technique has advantages and disadvantages, depending on the signal type and the analysis’s objectives. Of all the ways to diagnose rotating machinery failures, Machine Learning (ML) methods have gained the most relevance and growth recently [26,27,28,29,30]. Regardless of the algorithm used, all machine learning methodologies utilize a stage of information processing and reduction called Feature Extraction [31,32,33,34,35,36,37]. Feature extraction is a necessary digital signal processing (DSP) process to extract relevant information from a signal [38,39]. The goal is to reduce the dimensionality of the data and represent the signal in a smaller set of relevant features [40,41]. This process is often necessary as the original data may be poor quality, not in a format suitable for the algorithm, or contain redundant or irrelevant information.

Feature extraction can include techniques such as normalization, denoising, statistical value derivation, and one-hot coding, among others [42,43]. In fault diagnosis, feature extraction can reflect a change in machine components using DSP techniques and data analysis, such as vibration spectrum analysis, wavelet analysis, waveform analysis, and more [44,45]. These techniques transform condition-monitoring signals into a small set of features that a machine-learning algorithm can use to detect patterns and trends in signal values. There are mainly two ways of performing feature extraction: manual extraction of hand-crafted features and automatic extraction [46,47]. The use of hand-crafted features and automated feature extraction methods depends on different factors, such as the amount and complexity of the data, available resources, and the skills and time of the team. Despite advances in automated feature extraction methods, hand-crafted features remain valuable in fault diagnosis research for several reasons. Firstly, they offer high interpretability, allowing direct physical insights into fault mechanisms. Secondly, they provide flexibility in application across different types of rotating machinery. Thirdly, their computational efficiency makes them suitable for real-time monitoring applications. Lastly, hand-crafted features can capture domain-specific knowledge that might be overlooked by automated methods, potentially leading to more robust and generalizable fault diagnosis models. Automatic feature extraction is suitable when the data are extensive and complex, and there is no broad understanding of the phenomenon being investigated. However, the disadvantages of automatic feature extraction include complexity, difficult interpretability, and the need for an adequate volume of data [48,49,50]. On the other hand, hand-crafted features are more suitable when the data are small and the team has a broad understanding of the phenomenon being investigated. This methodology is more controllable and maintains simplicity in its implementation and execution, allowing a clear and physical interpretation in the context of the problem and a greater understanding of the features selected [51,52,53,54]. The final stage of the analysis and search for patterns for fault diagnosis is usually the classification task [55]. Fault classification is used to identify and categorize failures in a system. The goal is to classify an observation, typically a condition monitoring signal, into different fault categories, each representing a possible cause of failure. The features extracted from the signals are the inputs to the classification models, allowing machine learning algorithms to make a more precise separation between different failure categories [56,57,58]. The separability of the data for good class discrimination to occur by the various models and the interpretation of the classification results is directly related to the quality of the features extracted from the condition monitoring signals.

DSP-based hand-crafted features are created or designed to reveal and quantify specific information or behavior in the signals. Hand-crafted features can be extracted directly or indirectly by a human without the help of algorithms or automated tools. They may include methods such as manual calculation of statistical measures or definitions of specific knowledge about the phenomenon [42,54,59,60], in this case, the machine under study. Hand-crafted features can be more flexible and customized than automated methods. However, it can also be more labor-intensive and time-consuming in their search, design, implementation, function tests, and validations, especially in problems with many variables, for example, fault diagnosis of rotating machinery. The literature on these topics can become remarkably extensive and confusing since most focus on a specific application case and field of study. In Table 1, we show central reviews closer to manual feature extraction for condition monitoring on rotating machines. The table illustrates which main topics have been studied and how many Time features (TF), Frequency features (FF), Time-Frequency Features (T-FF), or Planetary Gearbox Features (PGF) have been mentioned or used for fault diagnosis purposes. Thus, highlighting a notable absence of a set of features.

For this reason, in the current work, we present an exhaustive compilation of hand-crafted features based on mathematical and statistical calculations. These features can be computed on condition monitoring signals for both time and frequency domains and used together as a classical feature extraction process. Each feature’s theoretical foundation is duly documented and presented as a mathematical formula.

This work aims to address several key research objectives: (1) To compile and standardize a comprehensive set of hand-crafted features from diverse fields for rotating machinery fault diagnosis; (2) To systematically evaluate the performance of these features across multiple databases and classification models; (3) To identify the most relevant and generalizable features for different types of rotating machines; (4) To assess the effectiveness of time-domain, frequency-domain, and fusion-based feature sets. By addressing these objectives, we seek to provide a thorough understanding of the role and effectiveness of hand-crafted features in modern fault diagnosis applications, supporting efforts to bridge the gap between traditional signal processing techniques and advanced machine learning approaches.

To address these research objectives, our study makes the following key contributions:A compilation of 169 hand-crafted features for condition monitoring signals, including 123 features for the time domain and 46 features for the frequency domain, based on a literature review of two decades. The features come from various fields and have mathematical/statistical foundations. This includes a unification of the nomenclature of the formulas and a categorization based on the aspect they are intended to measure.A rigorous evaluation of vibration signal features across seven feature ranking methods, three classification models, and seven datasets of vibration signals from gearboxes and bearings. One dataset is a public benchmark, while the others belong to our institution. Furthermore, features are evaluated under the same conditions.Analysis of top selected features by ranking methods across multiple datasets of different rotating machinery types. This demonstrates the effectiveness of classical feature extraction and provides insight into the most useful features for fault diagnosis in various mechanical systems.

The general restriction that the hand-crafted feature must maintain in our collection is that it must be self-contained, and its mathematical or statistical foundation must allow the calculation to be performed directly on the signal in any of the domains. There are many more features, some so sophisticated that for their calculation, they require the implementation of their complete algorithms, which, for the reasons previously stated, are not the objective of this compilation. It also does not consider types of signal processing that return the same signal but are modified, such as filters or TSA (Time Synchronous Averaging) [79]. Only simple hand-crafted features that seek to measure or quantify some specific phenomenon within the signal were collected, that is, that return a numerical value after the feature extraction process, converting the feature into a variable that can be analyzed. Finally, the results show high classification percentages for the various databases, showing that results comparable to deep learning methodologies can be obtained with adequate classical feature extraction. The evaluation shows that applying these hand-crafted features is helpful for any signal because they are calculated directly on them, and the vast majority do not have any configurable parameters, making the processing fast, optimizable, and with low computational cost.

The remainder of this paper is organized as follows: Section 1 provides a comprehensive compilation of hand-crafted features for classical feature extraction, including 123 time domain and 46 frequency domain features, along with their mathematical formulations and references. Section 2 describes the experimental data used for evaluating the performance of these features, detailing seven databases of rotating machinery faults. Section 3 outlines the proposed methodology for feature evaluation, including the feature extraction process, dimension reduction techniques, and classification techniques. Section 4 presents the results of the extensive experimentation, discussing the effectiveness of different feature subsets, ranking methods, and classifiers across various fault types and databases. This section also analyzes the most consistently useful features and their potential physical significance. Finally, Section 5 concludes the paper by summarizing the key findings, discussing the implications for fault diagnosis in rotating machinery, and suggesting directions for future research in this field.

## 2. Classical Feature Extraction

Condition monitoring is a systematic process that seeks to assess the current state of a system or component. In the case of rotating machinery, this may include vibration monitoring, which is the measurement of vibrations generated by the operation of the machine and the most widely used condition indicator for the diagnosis of machine faults [80,81,82]. Vibration is an oscillatory movement of an object. A vibration signal is a graphical or numerical representation of the vibration of an object recorded over time. The vibration signal can be measured with vibration sensors, such as accelerometers, and its analysis is performed using vibration analysis techniques, such as the Fourier transform or time domain analysis [83]. Vibration is an indirect way to measure machine health, as various factors such as wear, lubrication problems, structural problems, and other technical difficulties [84,85,86]. Consequently, these vibrations can be analyzed to detect possible machinery failures, known as vibration-based fault diagnosis. In fault diagnosis applications, feature values can be compared with predefined thresholds to determine normal or fault conditions, so selecting the appropriate feature is critical. In an ideal case, a significant feature is expected to distinguish normal conditions from fault conditions, establish a trend analysis, and avoid the influence of other equipment operating parameters [62,87]. However, the features are not necessarily informative for all cases or types of faults. For this reason, it is necessary to combine different types of features, so that information with the most remarkable possible diversity is obtained and can be used to reflect the machine’s condition, making it useful for fault diagnosis.

### 2.1. Signal Processing-Based Hand-Crafted Features

Feature extraction is a critical process in vibration analysis and fault diagnosis. It consists of identifying and selecting the signals relevant to fault diagnosis, which can be used to distinguish between different types of faults. These features can be extracted manually or automatically and may include measurements such as frequency, amplitude, and shape of the vibration signal. A direct way to achieve this extraction is by using digital signal processing (DSP) since one of its main tasks is extracting useful information and manipulating and transforming signals. The goal of using these DSP techniques is to find a new form of simple, effective, and reduced representation of the original signals. In the DSP, various transforms are used to analyze, manipulate, and represent signals differently. Some of the most common transforms include Laplace, Kalman, Wavelet, Hilbert, or Z transform. These are just some common transforms and techniques used in DSP. The choice of the appropriate transform depends on the signal’s characteristics and the problem’s specific needs to be solved because each method will reveal different information from the signal. In the present compilation, the search has been limited to using techniques that use simple Fourier transforms, such as the FFT and the PSD. This choice is because simple processes were sought and the possibility of reusing the transformation results in further processing. For these reasons, new representations of signals are sought, analyzed, and processed in the time or frequency domains. The time domain representation is the natural way a signal is presented and shows how the magnitude of a variable changes over time, whereas the frequency domain representation shows the energy distribution of a signal at different frequency components. Regardless of the domain used to represent and analyze a signal, a traditional way to extract information from it is through mathematical or statistical calculations. These calculations can be applied directly to the signal, using its data points as the statistic sample or input values to a function. Therefore, the computed values are then the features extracted from the signals, and a suitable set of features could efficiently represent a signal. This process is also a way to reduce the dimension of the data because now we use the feature set instead of the whole signal.

The information extracted from the signals will depend on the nature of the calculation or what is to be measured, quantified, or sought to be reflected with the features. In this way, we have some formulas (features) that measure the energy contained in the signal. In contrast, others seek to quantify the times the signal exceeds a certain threshold and many more types of information search. Thousands of processes, calculations, methodologies, and algorithms could be used to process a signal and extract some specific information from it. For instance, ref. [88] uses two approaches to align features in the time-frequency domain to classify faults. In addition, they use supervised and unsupervised techniques to improve the method. Besides, ref. [89] constructed a Wavelet auto-encoder, and then a deep approach to this model was performed. In [90], they apply a time-frequency transformation to express the signals. Then, a multi-scale TransFusion model is used to classify the features of time-frequency signals. Most groups of methods or techniques to extract features are applied in a specific way in different fields of application; the same ones seek all kinds of information and not one in particular. Many even depend on the specific application case where they will be used. For this reason, it is challenging to accurately categorize all of them based on the type of information they seek. To find the most significant number of hand-crafted features that can be calculated directly from the signal, an exhaustive compilation of formulas and calculations used as classical feature extraction methods on signals and applied in machine learning tasks has been carried out. These hand-crafted features have been chosen over the automatic ones mainly because, as an approximation to the DSP, they carry a direct physical significance with the phenomenon measured from the signals. This fact notably helps increase the ability to interpret the results within the context of the problem, in this case the diagnosis of failures in rotating machinery. We can also mention other significant advantages such as its simplicity, the control of the choice it offers the user, and its low computational cost in the execution. Other filtering criteria to accept features in this collection are that they are complete and self-contained, that their process can be represented in simple mathematical formulas, and that their calculation is conducted directly on the signal, whether in the time or frequency domain. Furthermore, we chose features that directly return real scalar numeric values after their computation and not processing styles that return the same but modified signal, such as filters, TSA, or noise cancellation techniques that are very common in DSP.

Features of various fields where signal processing is performed have been collected. Biomedical signal processing, such as Electrocardiography (ECG), Electromyography (EMG), or Electroencephalography (EEG), is a great exponent of fields where feature extraction is necessary. Other fields, such as audio signal analysis, roughness profiles, and entropy measurement, also require signal processing and are different from the field of fault diagnosis. Our compilation has shown that hand-crafted features follow certain trends in quantifying specific information they pursue according to the signal representation domain they use. In Figure 1, we present our proposed summary of the main information trends for which hand-crafted features try to quantify signals in the time and frequency domains. In this way, we can see that the information searched by the feature is closely linked to the domain to which it is applied. Signal analysis in different domains can help identify patterns and trends that are not evident in the original domain, which can be helpful in fault diagnosis, vibration analysis, anomaly detection, and other similar applications. The main trends in the time domain allude to the detection of patterns related to the waveform, while those in the frequency domain tend to search for changes in the frequency components of the spectra.

Some features search for or quantify information, as established in Figure 1, but do so in frames, bands, or segments of the signal in any domain. This is conducted to focus the search for information based on priori knowledge that a certain phenomenon should manifest in a specific signal locality, reducing the number of signal data points that would be used to calculate the feature. This would mainly reduce the “noise” of statistical features and make the calculation more relevant or discriminating. This happens more frequently with features from fields such as music or roughness since they seek to verify whether any variation manifests in specific parts of waveform or frequency spectrum segments. This hand-crafted feature calculation is mainly applied in the time domain when the signals are not stationary or their pattern varies over time, and in the frequency domain when it is known in advance that a band or frequency component should be altered for some reason. Because the literature reflects this type of emphasis in certain fields where feature extraction is applied, in the collection shown further, some features with the same formulation are counted as different, often depending on when applied to the complete signal or by bands or frames. As a substantial part of the contribution of this paper, a unification of the mathematical nomenclature has been carried out so that the calculation of the features and the information they quantify may be more understandable.

The unification of mathematical nomenclature presented in this paper is a significant contribution to the field. It addresses the inconsistencies in terminology across different domains and applications, providing a standardized framework for feature calculation and interpretation. This unified approach improves the clarity and comparability of feature extraction methods, facilitating better understanding and more effective use of these techniques in various disciplines.

For this reason, Table 2 shows a glossary of terminology. On many occasions, different authors use one version or another of the feature. However, they have different effects when classifying faults. This fact can be verified in the following sections, where the features are evaluated. The following subsections will detail the tables of the features collected for each domain.

### 2.2. Time Domain Features Recopilation

Computational analysis in the time domain is less expensive in terms of processing, and the only preprocessing required is signal conditioning. On several occasions, a visual inspection of various signal parts will be enough to detect abnormal behavior. However, there are non-stationary, chaotic, and noisy signals, such as vibration signals, where the detection of patterns visually is practically impossible. In those cases, it is essential to use features that help us to describe analytically the behavior of the signal. Time domain signal analysis is a natural technique for processing and analyzing the waveform evolution of signals over time, mainly because most sensors deliver measurements or waveforms over time.

Any technique used for feature extraction in the time domain will aim to analyze and characterize the waveform of the signal in terms of its temporal behavior. The choice of features to extract depends on the specific application and the type of signal to be analyzed. Examples of time-domain DSP techniques from which one or more features can be extracted and quantized include peak and valley detection, filtering, period inspection, amplitude and phase, distortion, denoising, and segmentation, such as identifying specific sections of the signal. Combining these techniques in a time-domain signal analysis process allows valuable information to be extracted and used for fault diagnosis, condition monitoring, and pattern detection in vibration signals. A complete collection of 123 handcrafted features in the time domain (T1 to T123) is presented in Table 3 to perform the classical feature extraction. The nomenclature of the formulas shown can be seen in Table 2, while references to the characteristics can be found in Table 4. The general trends of the information extracted by this type of feature are statistical values, waveform patterns, signal integration, entropy, event count, ratios, and hybrid values from the combination of several calculations.

### 2.3. Frequency Domain Features Recopilation

The representation of a signal in other domains allows us to analyze and understand its properties. For example, the transition from the time domain to the frequency domain allows us to analyze the signal in terms of the frequency and amplitude of its frequency components. This type of transformation can be performed using different techniques, such as the Fourier Transform, which converts a signal in time in its spectrum, representing the signal in the frequency domain. Analysis of signals in the frequency domain allows the identification of patterns and traits that are not evident in the time domain. This analysis is helpful for various applications, including the detection of abnormalities in the system that generates the signal, the identification of sources of interference and the identification of signal quality problems. In addition, it is also used in condition monitoring and fault diagnosis, as abnormal frequency patterns can indicate problems in mechanical, electromechanical, or electronic components of rotating machinery or systems. However, temporary information is lost in the transformation, which can be a disadvantage in some applications.

Based on the Fourier spectrum, other spectra analyses such as power spectral density (PSD) [91], bispectrum [92], trispectrum [93], and cepstrum [94], among others, have also been used in the diagnosis of rotating machinery faults. All the techniques mentioned above, excluding the frequency spectrum and PSD, are beyond the scope of this compilation study. These techniques involve more complex transformations that process or abstract the signal to various levels, some having multiple definitions or versions. The frequency spectrum represents the amplitude of the different frequencies that make up a signal. On the other hand, the spectral density or power spectrum is a graphical representation of the energy distribution in a signal as a function of frequency. When there is some alteration in a mechanical component, the frequency and power components also change; therefore, the position of the central spectrum peak will also change. This phenomenon can be reflected analytically using features that quantify those changes and reflect the condition of a machine. Table 5 presents 46 hand-crafted features in the frequency domain (F1 to F46). The nomenclature of the formulas shown can be seen in Table 2, while the references to the features can be found in Table 4. The general trends of the information extracted by this type of feature are values and statistical moments, the weighting of specific frequency components and ratios, and the search for values isolating harmonics or frames.

**Table 4 sensors-24-05400-t004:** References of collected features.

Features	Reference	Features	Reference	Features	Reference	Features	Reference
T1–T14	[44,63,95,96,97,98,99,100,101,102,103,104,105,106,107,108,109,110,111]	T29–T33	[97,98,99,100,101,103,112]	T65–T86	[113,114,115,116,117]	F1, F6, F7	[95,101,118,119]
T15–T17	[120]	T34–T36	[121,122,123]	T87–T107	[124]	F2–F4, F8–F12	[95]
T18–T20	[125,126]	T37–T41	[127]	T108	[128]	F5	[118,129]
T21–T22	[129,130,131,132,133,134]	T42–T45	[107,108,135,136,137]	T109	[138]	F13–F15	[129,137,139]
T23	[121]	T46–T53	[108,125,135]	T110–T113	[140]	F16–F25	[137,141]
T24–T25	[98]	T54	[44,101]	T114–T120	[129]	F26–F31, F40–F45, F47–F48	[129]
T26–T27	[122,123]	T55–T63	[126,136,137,142,143,144]	T121–T122	[129,130,131,132,133,134]	F32–F39	[67,138]
T28	[107]	T64	[107]	T123–T125	[67,138,145]	F46	[146]

The hand-crafted features in our study are designed to capture various aspects of vibration signals that are indicative of machinery conditions. In Figure 2, we illustrate a vibration signal in the time domain in which we exemplify the computation of features. In particular, this figure shows in yellow the computation of the T74 feature in which the magnitude between the third ridge and valley is identified. In purple, we see PPCM, which is the number of spaces between profile peaks crossing the midline to compute the Mean spacing in the mean line (Sm) or T76 feature. In navy blue, we identify the space between peaks SPB to compute the Mean spacing of adjacent peaks (MSBP) or T70 feature. In light blue, we show a region over a threshold (ROT), which is used to compute the High spot count (NROT) or T67 feature. We present other parameters as the number of peaks and valleys per *l* time frame, upper and lower thresholds, *N* that means the total number of times samples and *L* that means the total number of time frames as was established in Table 2. Naming the remaining parameters from features is vast, and it is not the purpose of this article.

These features can be broadly classified according to the signal characteristics they reflect:Amplitude-based features (e.g., peak value, crest factor): These reflect the magnitude of vibrations, which can indicate the severity of faults.Statistical features (e.g., mean, variance, skewness, kurtosis): These describe the distribution of vibration amplitudes, which can change with different fault types.Energy-based features (e.g., signal energy, entropy): These quantify the energy content and complexity of the signal, which often increase with fault severity.Time-series features (e.g., zero-crossing rate, autocorrelation): These reflect the temporal behavior of the signal, which can indicate periodicity or irregularities in vibrations.Frequency-based features (e.g., spectral centroid, frequency ratio): These capture the distribution of energy across different frequencies, helping identify characteristic fault frequencies.Shape-based features (e.g., impulse factor, margin factor): These describe the shape of the waveform, which can change with different fault types.

Each feature is sensitive to different aspects of the vibration signal, allowing for a multi-faceted analysis of machinery conditions. The combination of these features provides a comprehensive representation of the vibration signal, enabling effective fault diagnosis across various types of rotating machinery.

## 3. Experimentation Data for Hand-Crafted Features Performance Evaluation

The data employed to evaluate hand-crafted features and test their effectiveness in fault classification tasks add up to seven databases. The databases were obtained from different rotating machines, such as gearboxes and motors with shafts and bearings. Each database has its study elements with normal conditions and induced faults. The experimental configurations of the databases are diverse, have specific mechanical study elements, and reflect specific phenomena. An organizational scheme of all the rotating machines used, their derived databases, and study elements are shown in Figure 3. Some databases include multiple levels of fault severity in the elements under study. In contrast, others include various types of faults or fault combinations in their elements, known as multi-fault databases. Regardless of the specific phenomenon, machine, or element under investigation, these databases serve as valuable resources for conducting fault diagnosis studies. They are particularly useful because their data were collected using experimental setups designed to simulate various modes and effects of failures. As a result, these databases provide comprehensive, labeled information on different types of faults, making them ideal for research and analysis in the field of fault diagnosis.

The internal mechanical components of rotating machinery are diverse. However, some of these components are more prone to failure than others. Therefore, the most common gearbox faults that could appear in gears are tooth breakage, cracking, pitting, wear, chafing, and scuffing. Another type of fault is misalignment. Meanwhile, the main bearing fault is scratch and occurs mainly in the inner race, outer race, and rolling element. Another type of fault is eccentricity. In reciprocating compressors, failures occur mainly in valves and bearings. Figure 4 shows a general compilation of the main elements subject to failure of the different rotating machines and examples of the appearance of the associated fault. Another consideration is that six of the seven databases were produced by the Industrial Technology Research and Development Group (GIDTEC) of the Salesian Polytechnic University of Cuenca-Ecuador. For an external comparison option, the remaining database belongs to the Bearing Data Center of Case Western Reserve University, which is a famous benchmark used to test new techniques and methodologies for fault diagnosis.

The most relevant information about each database utilized to test the methodology followed in this study is briefly described below:**DB01:** This experiment involves a gearbox connected to an electric motor on the input shaft and an electromagnetic brake on the output shaft. The gearbox is a single-stage reduction type. The gears used are spur gears. Seven fault conditions are introduced: three broken tooth severities, misalignment condition, Pitting, and Pitting with face wear. More information on this database can be found at [147]. This database consists of vibration signals sampled at 1 Ks/s with a duration of 2 s, seven classes, and 150 observations for each class.**DB03:** This experiment involves a gearbox with ten health conditions, including chaffing tooth, worn tooth, broken tooth at three levels (25%, 50%, and 100%), 25% and 100% gear crack, 25% pinion broken tooth and 25% gear crack, 50% gear chaffing and normal condition in spur gears (number of teeth $Z_{1}=53$ and $Z_{2}=80$ with modulus 2.25 and impact angle 20°). The experiments were carried out with three loads, three constant speeds and three variable speeds, in each of them, five samples were collected in a duration of 10 s [148]. This database consists of vibration signals sampled at 50 Ks/s with ten classes and 90 observations for each class.**DB04:** The experimental setup consists of a gearbox connected to an electric motor on the input shaft and an electromagnetic brake on the output shaft. The gearbox is a two-stage reduction gear. The gears are of helical type. Ten different fault conditions are introduced, which include issues such as Scuffing, Pitting, and Crack in gears. In the case of bearings, the faulty components include the inner race, outer race, and rolling element. In addition, an eccentricity fault is introduced in the bearing housing. This database consists of vibration signals sampled at 50 Ks/s with a duration of 10 s, eleven classes, and 45 observations for each class [149,150].**DB05:** This experiment consists of nine severity levels of a broken pinion tooth. The gearbox is connected to an induction motor and an electromagnetic brake. The gearbox is a single-stage and is of the reduction gear type. The gears used are helical. The experiments were carried out with three loads produced by the brake on the gearbox’s output shaft. In each of them, five samples were collected for 10 s. Additionally, the experiment was performed at three constant speeds. This database consists of vibration signals sampled at 50 Ks/s with ten classes and 75 observations for each class [151].**DB08:** This is also a study of nine fault severity levels of the helical gearbox dataset. The gearbox is single-stage and is of reduction gear type. The experiments were carried out with three loads produced by the brake on the gearbox’s output shaft. The main difference between the DB05 and the DB08 is that the latter is more recent and captures more condition signals and observations. The experimental setup is the same, with differences in the materials of the elements and the procedures in the box assembly [152]. This database consists of vibration signals sampled at 50 Ks/s with a duration of 10 s, ten classes, and 180 observations for each class.**CWRU:** This database is a famous benchmark for testing bearing fault diagnostic techniques. Their page has information on the bearings used, the diameters and depths of the failures, and the configuration of the equipment assembly for the experiments. The experiment has four levels of speed and load. It has a normal class and induces failures in the outer race, the inner race, and the rolling element. Another important fact is that the bearing vibration signals were captured at a 12 kHz sampling rate. For the experiment in this paper, the 12 k fan end accelerometer data were used [153]. This database consists of vibration signals with a duration of 0.1667 s, four classes, and 80 observations for each class.**DB02:** This dataset aims to study bearing behavior. It involves a *ϕ*30 mm shaft with flywheels mounted. The shaft is connected to an induction motor and is seated on the bearings and their housings. The acquired signals belong to an accelerometer placed in a vertical position near the movement source. Details of the experimental setup are described in [151]. This database consists of vibration signals sampled at 50 Ks/s with a duration of 20 s, seven classes, and 45 observations for each class.

Accelerometers are strategically placed closest to the source of motion on the machine across all databases to capture data effectively. All signals from all experiments were stored as a group in a database, differentiating according to the failure labels they contained. The condition monitoring signals used for this experiment have been limited to vibration signals since vibration analysis has traditionally been the most widely used means of detecting anomalies in rotating machines. Especially when combined with spectral analysis, it provides a direct association of the characteristic frequency components that a machine should have based on the movement or rotation of the elements that compose it. Figure 5 shows a compilation of vibration signals taken from the normal condition of each database used, compared to their respective frequency spectrum. When analyzing the vibration signals over time, it can be verified that certain signals present a slightly more cyclical behavior, as could be the case for DB03 and DB05. On the other hand, the vibration signatures of the other databases seem much more chaotic. On the other side of the frequency spectrum, one can observe clear differences in their different frequency components. The DB02 and the CWRU clearly show characteristic signatures with frequency sidebands, while the other spectra show that most of their components are housed in the lower frequencies. Clear differences are distinguished between the different time and frequency graphs among all databases. These differences emphasize that each machine has many internal behaviors that must be characterized to correctly diagnose a fault.

## 4. Proposed Feature Evaluation Methodology

After the compilation of hand-crafted features for their application in condition monitoring signals, a systematic methodology is proposed to establish an adequate reference framework to carry out the exhaustive feature evaluation in fault classification tasks in rotating machinery. Figure 6 shows graphically the sequence of steps followed in the proposed methodology.

**Data acquisition and adaptation:** Different vibration signals were collected from seven databases and stored separately. The signals are organized in local directories, separating the normal condition and its different fault modes for each database. Seven original datasets with vibration signals were acquired from different machines with different fault configurations, number of observations, acquisition rates, and signal sizes.**Feature extraction step:** In this phase, the calculations of the hand-crafted features were applied according to the mathematical formulations shown in Table 3 and Table 5. The 123 time-domain and 46 frequency-domain hand-crafted features were implemented as functions of the MATLAB software R2023b, and an algorithm was created that is responsible for iterating them for each signal from all databases. To exhaustively evaluate the features, they were organized into three large evaluation groups: one with only time domain features, another with only frequency domain features, and the last group containing a data fusion scheme. The data fusion consists of concatenating time domain and frequency domain features into a single-dimensional feature vector for each signal. Thus, Group 1 evaluates 123 features, Group 2 evaluates 46 features, and Group 3 evaluates 169 features. Each group contains separately the calculated hand-crafted features for all databases. Subsequently, the features were organized in a matrix format for the later machine learning process, obtaining 21 data corpora, seven per evaluation group corresponding to one per original dataset.**Feature selection:** Features were normalized and filtered by correlation. Subsequently, seven feature ranking methods are applied to each of the 21 data corpora to obtain the ten most important features of each one. The feature ranking methods used in this feature selection step were as follows: RelieF Algorithm [154] (RA), Chi-Square [155] (CS), Information Gain [156] (IG), Pearson Correlation [157] (PC), Fisher Score [158] (FS), Gain Ratio [159] (GR), and Random Forest [160] (RF). A total of 147 data subsets were generated, i.e., 49 subsets per evaluation group, based on the ten most relevant features. Part of the exhaustive evaluation of the hand-crafted features is also focused on knowing their relative importance within the datasets used to classify rotating machinery faults, measuring the individual contribution of each one, and observing the relative persistence of each feature. For this reason, several ranking methods were used to enrich this process. Each ranking method has different mathematical foundations, as do the calculated features, which would allow valuable conclusions to be drawn in case of finding patterns within the top 10 lists of selected features. We have used the ranking methods that we consider to be the most important and reported in the literature.These methods were selected based on their use in mechanical systems. For researchers seeking a more comprehensive understanding of each approach, references to seminal works are provided.**Classification step:** In this final phase, each of the 147 data subsets was evaluated with three different machine learning classification models: Support Vector Machines [161] (SVM), k-Nearest Neighbors [162] (KNN), and Random Forest [163] (RF). The results were compared in terms of the best performance achieved (accuracy percentage) and the number of features needed to reach that value. As in the previous step of feature selection, in this last step, we try to enrich the evaluation process using three classification models. As previously mentioned, each feature and ranking method has its mathematical foundation, and in the same way, it happens with the classifiers. In this way, we have a parametric classifier, one based on proximity and one based on trees. The diversity of classifiers and ranking methods contributes to a comprehensive feature evaluation. The aim is to compare the classification results and find trends with the values achieved, the feature groups used in the process, and the frequency of appearance of certain features in the top 10 list.

An additional fact to mention is that a standard and fixed configuration was used in the configuration of hyperparameters used by the classifiers and ranking methods. Therefore, SVM uses a linear kernel with a penalty of 10, KNN uses k = 3 with an s-euclidean distance metric, while RF uses a tree number of 40. All classifiers were trained following the five folds (k-folds) validation schema. No hyperparameter search, optimization step, or process is performed because the methodology seeks to evaluate the feature’s ability to separate classes and not the power of the classification model. The developed methodology has been applied transversally to different types of rotating machines. With the successive application of these steps, this methodology is expected to allow a more in-depth understanding of fault patterns and a systematic feature evaluation to validate their use in fault diagnosis tasks. The summary of results is grouped according to the mechanical components under study from the different databases. In this way, Table 6 shows the result condensation of the databases that use gearboxes with spur gears, Table 7 shows the results for gearboxes with helical gears, and Table 8 shows the results for bearings. The maximum classification percentage (Cl. %) achieved and its number of features (#F.) are shown.

The total summarized process of the methodology followed for the exhaustive evaluation of hand-crafted features can be seen in Figure 6. In this way, the different domains in which the features were calculated could be evaluated in an isolated manner, as well as their potential and contribution as a whole. The training was also carried out exhaustively. First, the most important feature is taken from a subset of data containing the ten best features ranked by some method. Then, the three classifiers are trained to obtain classification performances. The next step is to take the first two most significant features and retrain all three classifiers. This process is repeated iteratively, adding the next most important feature according to the ranking method in each iteration until a total of 10 features are reached to carry out the training. This process is repeated for the 147 data subsets, constantly training the three classifiers. This process is conducted to find the best result (highest classification performance) by a ranking-classifier combination, the number of features required to achieve said performance, and the individual contribution of each feature. The summaries of these results with the best and worst results can be viewed in Table 9 and Table 10.

## 5. Results and Discussion

This section presents the results obtained from the experimentation to evaluate the performance of different hand-crafted features in fault classification tasks over rotating machinery. A total of 147 subsets of the top 10 most important features were obtained by applying seven ranking methods to seven original databases with three evaluation groups. Each of these data subsets was exhaustively evaluated by three different classification models. In the first place, the general results of the experiment are presented and organized according to the mechanical components associated with the different databases, namely spur gear gearboxes, helical gear gearboxes, and rotating machinery bearings. The tables show the percentage of classification and the number of features needed to achieve that value. Additionally, the results are condensed by database, ranking methods, and evaluation group, allowing a direct comparison and identifying the combinations that provide the best and the worst classification results. In this way, Table 6 summarizes the results obtained in the three evaluation groups for the DB01 and DB03 databases, the same ones associated with gearboxes with spur gears. Table 7 shows the overall results obtained for the databases associated with gearboxes with helical gears, specifically DB04, DB05, and DB08. For bearings, Table 8 displays the results corresponding to the DB02 database and the external benchmark CWRU. The classification results are shown in Table 6, Table 7 and Table 8. They show that ranking methods significantly impact the effectiveness of fault diagnosis. For example, suppose that a ranking method-classifier combination shows a high classification percentage and a low number of features. In that case, that method is very effective in identifying the most critical features. Furthermore, if a particular database shows a high classification percentage with a particular ranking method, this would indicate that the method is especially effective in identifying essential features for that specific database. In this way, the fault diagnosis can be performed with fewer data and greater precision.

Table 6, Table 7 and Table 8 also show the results by evaluation groups. That is, it shows the classification results and the number of features associated with Group 1, in which only the features of the time domain were used in its evaluation (123 in total). For Group 2, they used only the frequency domain characteristics (46 in total). Group 3 uses a data fusion scheme, combining time and frequency features (169 in total). It is essential to highlight that the results reflect the capacity of the classifiers to carry out fault diagnoses in rotating machinery and the effectiveness and usefulness of hand-crafted features calculated in the time and frequency domains. It should also be mentioned that if we read the results of Table 6, Table 7 and Table 8 by rows within each evaluation group, the data subset used is the same and will produce different performances with each classifier. The set of features used varies when reading the results between rows since a different ranking method will choose a different set of features. Even with this, one or several features may be repeated in different evaluation groups or data subsets that were selected by different ranking methods. A high percentage of classification and a low number of features indicate good feature selection. Therefore, combining ranking methods with hand-crafted features is a viable way to perform fault diagnosis in various rotating machines. The next part of the results obtained refers to the best and worst results obtained from all the exhaustive experimentation. In this way, Table 9 shows the best classification results obtained for each database with the respective set of features used, the deviation of the precision in the training, and the individual contribution of each feature within that configuration. Similarly, Table 10 shows the same information with the difference that presents the worst results obtained in all experiments. Both tables highlight in bold the maximum precision obtained by the best-performing classifier. It can be seen in both cases that, on some occasions, the classifier reaches its maximum precision with a certain number of features and that the act of adding more is counterproductive to the overall performance of the model. Table 9 shows that the evaluation groups that produced the best features were Group 1 of time alone and Group 3 of data fusion. It can also be seen that the ranking method with the best feature subsets is Random Forest, followed by the Relief Algorithm. The vast majority of the contribution to accuracy is achieved by using the four most important features.

For DB01, the best resulting combination is the RF method and the RF classifier, reaching 87.09% with eight exclusive features of the time domain. For DB03, the best result is achieved again with a set of time-exclusive features in the combination of the IG method and the KNN classifier, reaching 98.77% with 10 features. In DB04, groups 1 and 3 achieved equal precision of 98.59% using the KNN classifier. However, group 1 used the RF (Random Forest) ranking method, while group 3 employed the CS (Chi-square) ranking method. The only substantial difference between these results is that the time-exclusive feature group achieves this percentage using only eight features, whereas the fusion group achieves it using nine features. DB05 has a tie again, this time within the same group, with the exclusive time features being the winners. A maximum classification percentage of 93.07% was achieved. The first combination was the RA method and the RF classifier using 10 features and an acc_std of 2.14. In comparison, the second combination was the RF method and the KNN classifier using seven features and an acc_std of 3.42. The relatively high dispersion in precision values in the second combination could be attributed to the features used. Therefore, the table will include the list of the 10 features from the first winning combination. DB08 has a tie between the time-only and fusion groups. In both cases, an accuracy of 99% is achieved with 10 features with a combination of the RF method and the KNN classifier. In the first case, using the time-only group, an acc_std of 1.01 was achieved, while in the second case, employing the fusion group, an acc_std of 0.54 was obtained. A detail to highlight is that in the time group feature list, the features of Hr, NPP, and meanInflectPoints appear in all the sets of features for all previous databases. The last three databases are for helical gear gearboxes, and the five databases analyzed so far are for gearboxes. For DB02, the maximum classification percentage of 92.66% with the combination RA method and the RF classifier with an acc_std of 2.67 requires 10 features to reach this value. Finally, in the case of the external CWRU database, an amalgamation of almost perfect results is obtained. The maximum classification percentage achieved is 100% with the combination RF method and the KNN classifier using only two features of the fusion group, with an acc_std of 0. In this case, many combinations are obtained that return excellent results for the three groups of assessment. The fusion group has been chosen as the winner for the CWRU because it is the one that obtains a combination that requires fewer features to achieve the best results. It is worth mentioning that, while it is true that a maximum result was achieved in the fusion group, the features used to achieve this level of classification are features that are calculated in the frequency domain. The total list of features that appear in this subset is placed in Table 9. In this table, DB01 contains the first five features, which are associated with the counting, statistical, hybrid and ratio, and shape and profile features types. In DB03, the first five features are associated with the statistical, shape and profile, and hybrid and ratio features types. In DB04, the first five features are associated with the hybrid and ratio, counting, and shape and profile features types. In DB05, the first five features are associated with shape and profile, hybrid and ratio, statistical, and counting feature types. In DB8, the first five features are associated with hybrid and ratio, counting, and shape and profile feature types. In DB02, the first five features are associated with counting, central tendency and statistical moments, statistical, and hybrid and ratio feature types. In the CWRU database, the first five features are statistical, and hybrid and ratio features.

On the other hand, reviewing Table 10, it can be seen that, in general, the frequency-only group has the worst overall performance in precision with the classifiers, although acceptable values are reached for certain databases. It can also be seen that the classifier with the worst global results is the SVM. This could be because a linear kernel was used in its configuration, and not all classes within the databases have to be linearly separable. This could also indicate the complexity of the data available to diagnose failures. We must consider the results obtained in Table 9 with other essential results to obtain a complete view of the experiment. We have created distributions and ratios of the occurrence graphs of the evaluation groups that appear in Table 9 only, since they are the groups that appear in the best results, that is, time-only and fusion groups. The bar charts in Figure 7 and Figure 8 show the frequency of occurrence of features in the 49 data subsets with the ten most important features per only time and fusion evaluation groups, respectively. The graphs show features such as hr, NPP, WF, skewness, meanInflectPoints, rc, mean, SC, FR, and SFP, among others, as the features with the most occurrences or repetitions in the two evaluation groups. In the case of the only time group, Figure 7 shows how the hr feature has the most occurrences, totaling 30 out of 49. This figure also shows two more features, such as NPP and WF, that follow hr as the most used features in this experimentation group. The data fusion group shares many features that appear in Group 1, but it also includes certain features that are calculated in the frequency domain and have notorious appearance levels. Figure 8 also shows features such as FR, SFP, and SC that appear more than 15 of the 49 times. As a curious example, we have the SC feature, a version of the spectral centroid computed over frames of the power spectrum by averaging the power-weighted frequencies within the frame of the spectrum. This feature combines the trends of calculations based on frequency weighting with those seeking information in sections of the spectrum, as discussed in section II. The hr feature is another example of a calculation over frames, and it belongs to the time domain. In contrast, features such as FR and NPP are calculated over the whole signal in their respective domains.

To accompany the feature occurrence distribution graphs, Figure 9 and Figure 10 are shown, summarizing the features that appear the most times in the ranking lists for evaluation groups 1 and 3, respectively. These graphs show those features that appeared in more than three ranking lists. That is, more than three ranking methods put them somewhere in their top 10 most significant features. Virtually, the maximum value a feature could reach in these graphs is 7 since there is a maximum of seven ranking methods. However, for features that apply their calculation in frames or bands, there are cases where the same feature, calculated in different bands, appears several times in the same list of the top 10 most relevant features by some ranking method. This can be seen clearly with many examples in Table 10, and we have the Env and SRF features cases in Figure 9 and Figure 10, respectively.

Analyzing Figure 7, Figure 8, Figure 9 and Figure 10 together, we can see that some features appear more frequently than others. The count distribution shown in the bar graph in Figure 7 suggests that the features with the highest occurrences could be considered the most important in the evaluation performance experiment for Group 1. We can confirm this by looking at Figure 9 and Figure 11 alongside Table 9. Taking DB03 as an example, Figure 11 shows that the hr feature appears in the first position of the rankings 29% of the time, and we already know from Figure 7 that this feature appears the most times in the time-only evaluation group. Inspecting Figure 9, we can see that the hr feature is chosen by six of the seven ranking methods used in the experiment. Finally, we can see in Table 9 that the best classification result for DB03 is achieved with the hr feature in the fifth place of importance. Additionally, even in Table 10 of the worst results, we can find the hr feature in the first position of importance in the configuration shown for DB03. Several other features follow this pattern of occurrences and are important in fault classification as hr. We have examples such as the WF for DB04 or the NPP for DB01 in the time-only group, while we see cases such as stda for the CWRU in the fusion group. There are also cases where a feature appears many times in a few ranking methods but is located in significant positions, as is the case of the FR feature for DB08 in the fusion group. In this example, the FR feature does not appear in Figure 10 but stands out in Figure 12 and Figure 8, in addition to demonstrating its importance in the results of Table 9.

The features with the most occurrences or selected by various ranking methods seem to show a great association with the study phenomenon. Therefore, they are the most useful in classification processes. Furthermore, the frequency of appearance of each feature in the top 10 lists may indicate the consistency of the ranking methods used to evaluate the performance of handcrafted features. If a feature frequently appears in the top 10 lists, it is evidence that it is considered essential by various ranking methods. On the other hand, if a feature appears infrequently on the top 10 lists, this suggests that it is considered less important or that the ranking methods used in the assessment need to be more consistent in their assessment of that feature.Figure 8 and Figure 9 are valuable tools for understanding the relative importance of each feature in diagnosing rotating machinery faults. To better understand the relationship between the performance of the classifiers, the databases, and the features selected by the ranking methods, we must observe the results in Table 9 and Table 10 with Figure 11, Figure 12, Figure 13 and Figure 14. The sunburst graphs in Figure 11 and Figure 12 show the top four critical features per database and their percentage of occurrences. Those graphs have been built using the results obtained from the first four places of the 49 lists of the top 10 most important features for each evaluation group and directly associating them with the databases to which they correspond. This type of visualization allows for a clear representation and an easy way to understand the data hierarchy, allowing us to see which features have a more significant influence within each of the databases and how they are related to each other. The fact that some features have high percentages of appearance in certain ranking positions within a database indicates that those features are critical for diagnosing failures in that specific rotating machine. This relation means that the repeatability in the feature ranking has allowed us to identify the most relevant features for each database and, therefore, focus efforts on understanding those specific features to detect faults efficiently and effectively.

Figure 13 and Figure 14 present the other sunburst graphs that show the percentage of appearances of a feature in any position of the top four of importance, associated with the percentage of times that a ranking method places it in that particular position. The ring-shaped representation allows for a global vision of the performance of the features in the rankings and their ability to be identified as necessary by the different ranking methods used in the experimentation. It is essential to consider that the percentages of appearances of a particular feature in a position of the top 4 are global for all the experimentation; that is, the percentage of appearances marked by a feature is, in general, for all the databases used in the experiment for each evaluation group. If a feature has a high percentage of occurrences in some position in the top 4, it means that the ranking methods frequently identify it as an important feature in that position, making it valuable in a horizontal way; that is, its use is viable for many types of rotating machinery for fault diagnosis. Figure 13 shows interesting information, such as that the WF and NPP features have the highest occurrence percentages as the most important features with 16% and 10%, respectively, for the time-only evaluation group. On the other hand, Figure 14 shows the features of mean and WF as those with the highest percentage of appearance in the first place of importance in the fusion evaluation group, being chosen by almost all ranking methods at least once. If we analyze those graphs together with Figure 7 and Figure 8, we can see that all the aforementioned features appear several times in their respective experimental groups, which suggests that this enhances the possibility of ever finding them at the top of the podium of the feature importance.

It should be noted that the bar graphs in Figure 7 and Figure 8 and the sunburst graphs in Figure 11, Figure 12, Figure 13 and Figure 14, only show the results of evaluation groups 1 and 3; the time-only and data-fusion feature sets. This election, as stated previously, occurs because these groups are present in Table 9 with the best performance in the classification task, so more significant associations can be derived between features and databases. If we try to detect patterns in the results obtained, we can find certain coincidences with different databases with similar study elements. For example, the hr feature is present with different percentages in DB01, DB03, DB04, DB05, and DB08 in different places of importance. Furthermore, all these databases study gearboxes with helical and spur gears, and the hr feature appears in four of the five sets of features used that achieved the best classification results, as we can see in Table 9. We also have DB02 and CRWU, which have results that share features such as Mean and SFP. Another common trend that we can observe in the features of the top 4 organized by the database they represent is that they usually exchange the position in which they appear within the ranking of the top 4. A high percentage of occurrences in a given top 4 position means that the feature is important and valuable in various fault diagnosis application cases in different types of rotating machinery. In addition, this also indicates that the ranking methods are consistent in identifying said feature as one of the most important. Associating the results shown in Figure 11, Figure 12, Figure 13 and Figure 14, and Table 9, we can identify that, for a specific database, the ranking method with the best performance places a feature in a high position in the top 4 with a high percentage of appearances, and that the classifiers used in that database achieve a high percentage of classification with some essential features. This result is excellent evidence that this feature can be very informative and may suggest good physical significance for diagnosing faults in that particular database and other similar ones that also capture condition information on rotating machine processes.

## 6. Conclusions

This study presented a comprehensive compilation of 169 hand-crafted features for fault diagnosis in rotating machinery and evaluated their effectiveness using seven databases and various classification models. Our findings demonstrate that hand-crafted features can achieve high classification accuracy, up to 99% in some cases, for various types of rotating machinery faults. Furthermore, the combination of time and frequency domain features often outperformed single-domain feature sets. The advantage of hand-crafted feature extraction is high interpretability, allowing for direct physical insights into fault mechanisms. These features also offer flexibility in application across different types of rotating machinery and show comparable performance to more complex methods, including deep learning approaches.

In this study, a series of formulas and calculations, simple methods, were collected from the literature to perform hand-crafted feature extraction on signals in the time and frequency domains. The proposed methodology for evaluating the performance of hand-crafted features in fault classification tasks has proven to be practical and valuable. Furthermore, the results strongly suggest that it is possible to achieve a good fault diagnosis performance using a combination of different ranking methods and calculated hand-crafted features. The results obtained show the importance of analyzing different domains to achieve a good diagnosis of faults in rotating machinery. The proposed methodology allowed us to objectively compare the performance results of the features, and the ranking methods used allowed us to identify the most relevant features in each database. Some features have a high percentage of occurrences anywhere in the top 4 of importance. They are valuable and viable for various fault diagnosis applications across many types of rotating machinery. In addition, simplicity and low computational cost are some advantages of using hand-crafted features for classical feature extraction processes in condition monitoring signals since, among other things, they allow a whole series of experiments and tests exhaustively without excessive time consumption.

A relative repetitive pattern is seen between relevant features in some databases that have mechanical elements in common. This could suggest a direct physical relation of the phenomenon with the feature values. Despite this, additional analysis and testing are needed to fully confirm and establish the physical significance between the feature and the physical process or mechanical component. The results suggest the importance of a methodical and systematic approach to analyze characteristics and evaluate the performance of features through the classifiers in diagnosing faults in rotating machinery. The classification percentages and the number of features required to achieve that value were informative in determining the effectiveness of the features and identifying those that may be useful for fault identification. These results also suggest that it is crucial to consider interpretability and interpretation of features, as this can significantly impact diagnostic effectiveness. In the same way, the compilation of features carried out and presented in the summary tables is a substantial part of the contribution of this paper. Due to the disparate nature of the literature in terms of the mathematical formalization that the features have, we have found it necessary to unify the nomenclature of the formulas in this compilation. The same summary of nomenclature can be seen in Table 1. A map was built that organizes and specifies in which domains the different types of features are used. The main trends of the features in terms of the information they seek to reveal of the signal were identified. The fact that these diversified trends exist according to the signal representation domain makes sense since each domain has its advantages, disadvantages, and specific objectives of what they want to reveal about the signal; therefore, the features will seek to exploit precisely those strengths, such as waveforms in time and energy distribution in frequency components in the spectrum.

Assuming that the acquired signals have a sufficient sampling rate to effectively capture the phenomenon being measured and are not affected by aliasing or noise, another advantage of hand-crafted features is agnostic to the acquisition rate and time. Whether the signals are large or small, fast or slow, feature calculation on any signal will always be viable, and there is virtually no restriction on the size of the input to the processing since the information extraction stage is separated from the inference part, which does not happen in Deep Learning models [164,165]. This advantage applies particularly to those signals in the time domain. Some exceptions could cause errors, such as counting the number of zero crossings of a time domain signal with no negative side. The result will be 0, which is not a significant feature in this example. On the other hand, the features in the frequency domain could become dependent on the sampling rate due to its direct relationship with the resolution of the spectrum. If the size of the sample corresponding to the signal is reduced, a decrease in the performance of the features in frequency is also expected. Consequently, the precision of the classifiers that use them will decrease. The choice of which features or calculations to use for each case will be left to the judgment and expertise of the subject who performs the analysis. In this work, databases that had signals with different sampling rates and acquisition times were used, achieving very high classification results using hand-crafted features without any problem. In addition, standard methods and models were also used for ranking and classifiers. There was no hyperparameter optimization, so many combinations of rank classifiers could significantly improve their performance if this other process was applied. This election is because the experiment aimed to measure the efficiency of the features, not the power of the classification model. If a group of characteristics serves to classify failures in a rotating machine, then these features are very informative to reflect the status or type of failure of a particular component; therefore, the fault associated with a component of the machine has been characterized. In subsequent actions within a CBM plan, these features and their evolution can be used as markers that provide information on the deterioration of the machine or any of its components. In this way, we would get closer to performing predictive maintenance.

The yields obtained in the classification are comparable with many others based on much more sophisticated processing or with methods based on deep learning with the advantage that they are interpretable and computationally very light. It is necessary to mention that there is no such thing as an optimal group of features that serve all databases. Instead, there is a pool of features from which some can be extracted to achieve acceptable performances in fault classification tasks for rotating machinery. Part of these pools would be in the sunburst of occurrences since the graphs show all those features that were chosen by more than three different ranking methods. Furthermore, this work has found some specific features that work for many datasets that use the same mechanical components. It is also necessary to evaluate other databases of rotating machinery, other methods of dimension reduction in the data, and aspects such as generalization and robustness to variations in the condition monitoring signals to provide further completeness in feature evaluation. It is necessary to continue researching and developing new techniques to achieve an even more precise, effective, and generalizable diagnosis in the future and optimize the feature’s interpretability. Although favorable results were obtained, it is important to stress that research is still needed. In future research, it would be interesting to explore the combination of automatic and hand-crafted feature extraction methods that involve other domains or signal transformations or even further expand the feature collection to achieve better performance in rotating machinery fault classification tasks.

Future work will focus on comparing the performance of these hand-crafted features against automated feature extraction methods, particularly deep learning approaches. This comparison should include an expanded range of classification models, incorporating both traditional machine learning and deep learning architectures. Such a comprehensive evaluation would provide valuable insights into the relative strengths and limitations of manual versus automated techniques for fault diagnosis in rotating machinery, potentially revealing new perspectives on feature effectiveness across diverse learning paradigms.

## Figures and Tables

**Figure 1 sensors-24-05400-f001:**
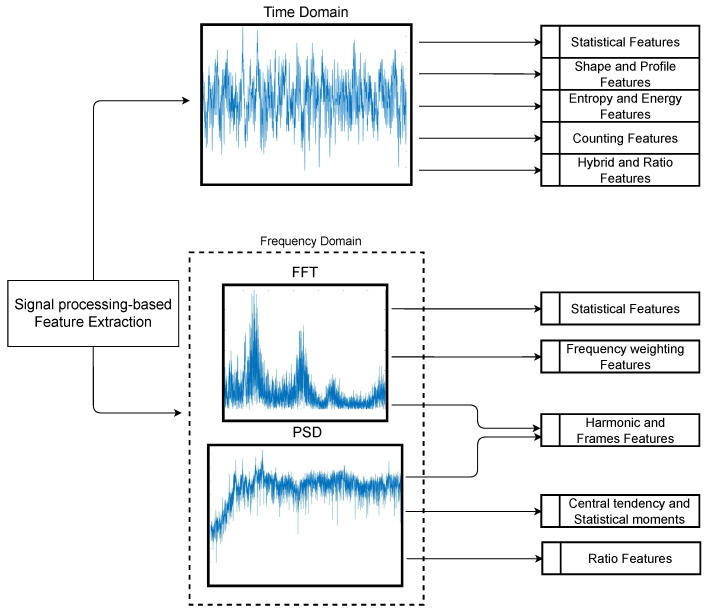
Main trends in information search for hand-crafted signal processing-based features in the time and frequency domain.

**Figure 2 sensors-24-05400-f002:**
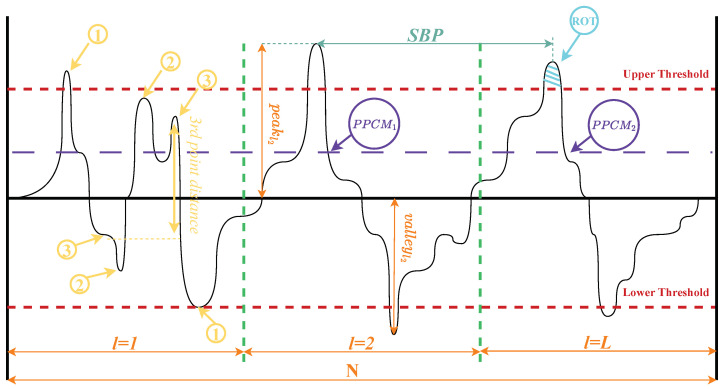
Illustration of some feature parameters on a vibration signal.

**Figure 3 sensors-24-05400-f003:**
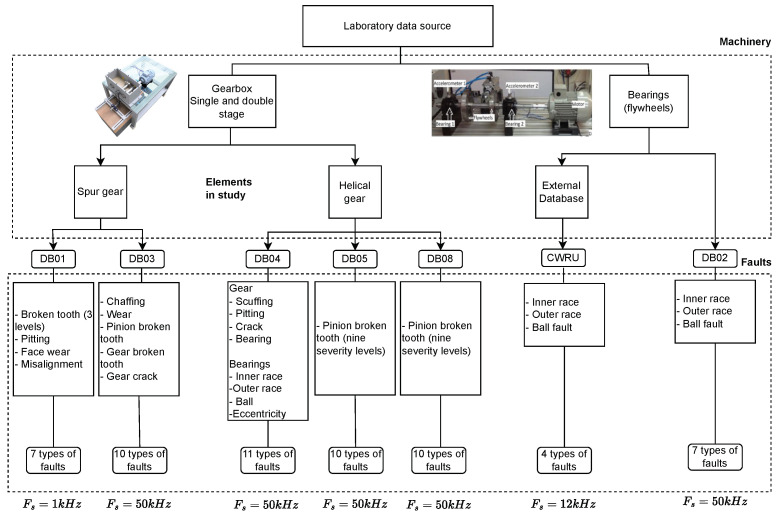
Schematic description of the databases used.

**Figure 4 sensors-24-05400-f004:**
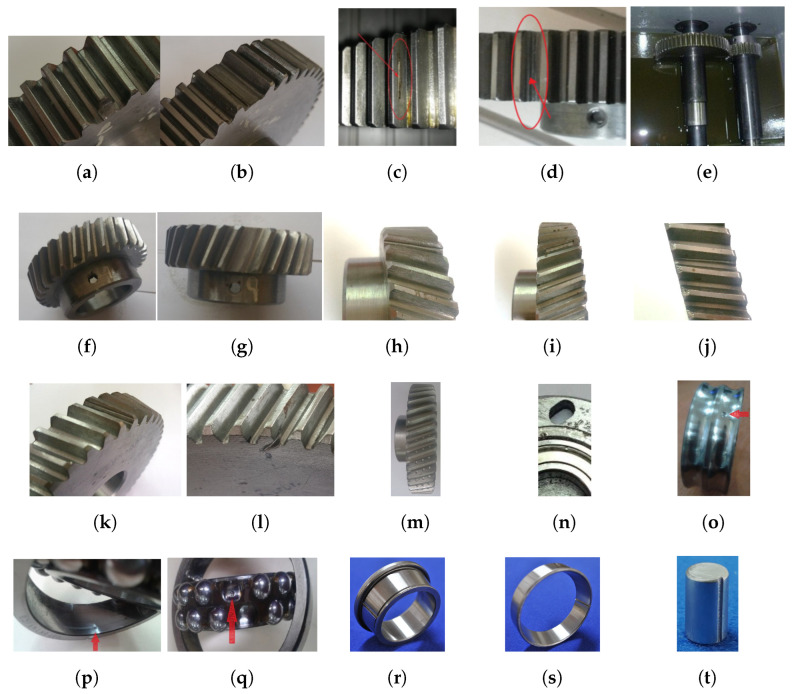
Compilation of the main mechanical components prone to failures under study in the different databases. (**a**) Broken tooth 25%. (**b**) Broken tooth 100%. (**c**) Scuffing. (**d**) Wear. (**e**) Misalignment. (**f**) Broken tooth 11.5%. (**g**) Broken tooth 100%. (**h**) Scuffing. (**i**) Scuffing 10 mm large. (**j**) Scuffing one stripe. (**k**) Scuffing two stripes. (**l**) Crack. (**m**) Pitting. (**n**) Eccentricity. (**o**) Inner race fault. (**p**) Outer race fault. (**q**) Rolling element fault. (**r**) Inner race fault. (**s**) Outer race fault. (**t**) Rolling element fault.

**Figure 5 sensors-24-05400-f005:**
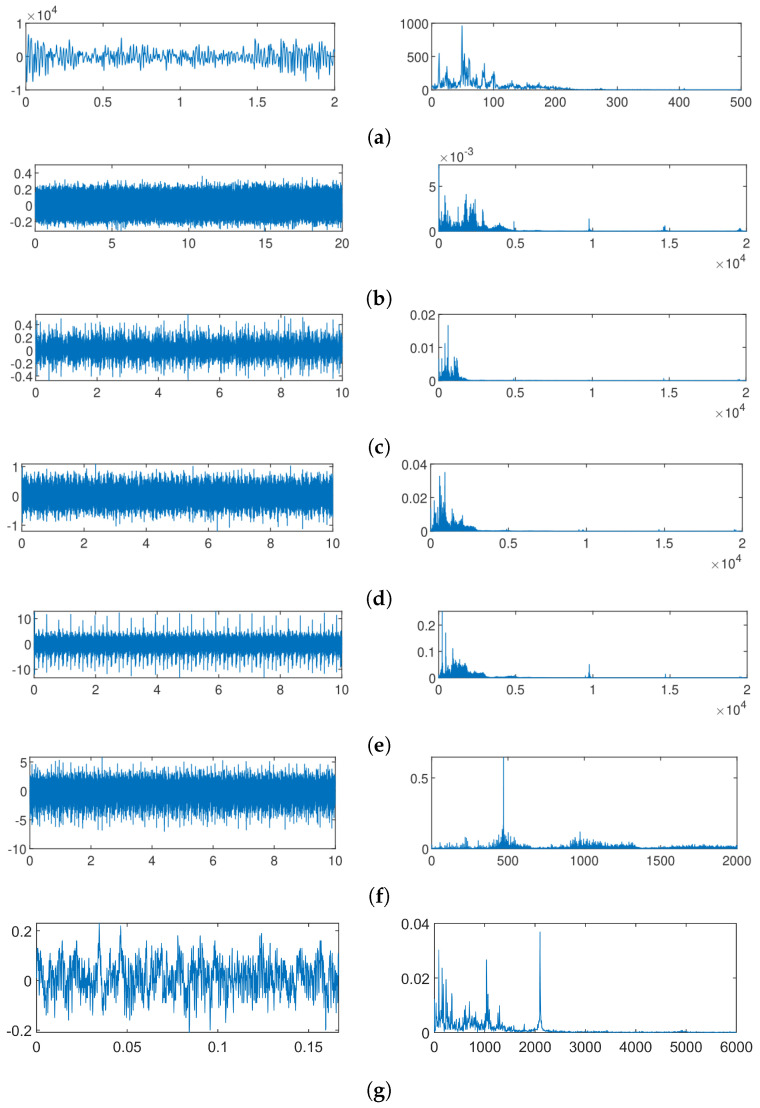
Example of a time domain signal alongside their respective frequency spectrum per each database. (**a**) DB01. (**b**) DB02. (**c**) DB03. (**d**) DB04. (**e**) DB05. (**f**) DB08. (**g**) CWRU.

**Figure 6 sensors-24-05400-f006:**
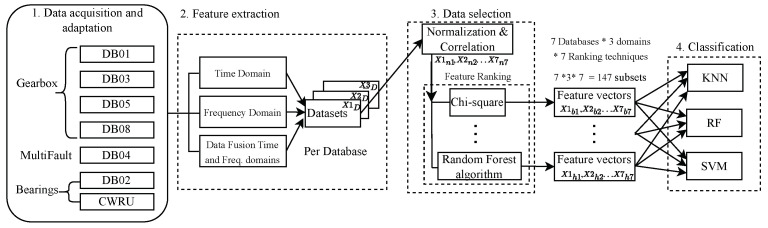
Diagram of the methodology employed.

**Figure 7 sensors-24-05400-f007:**
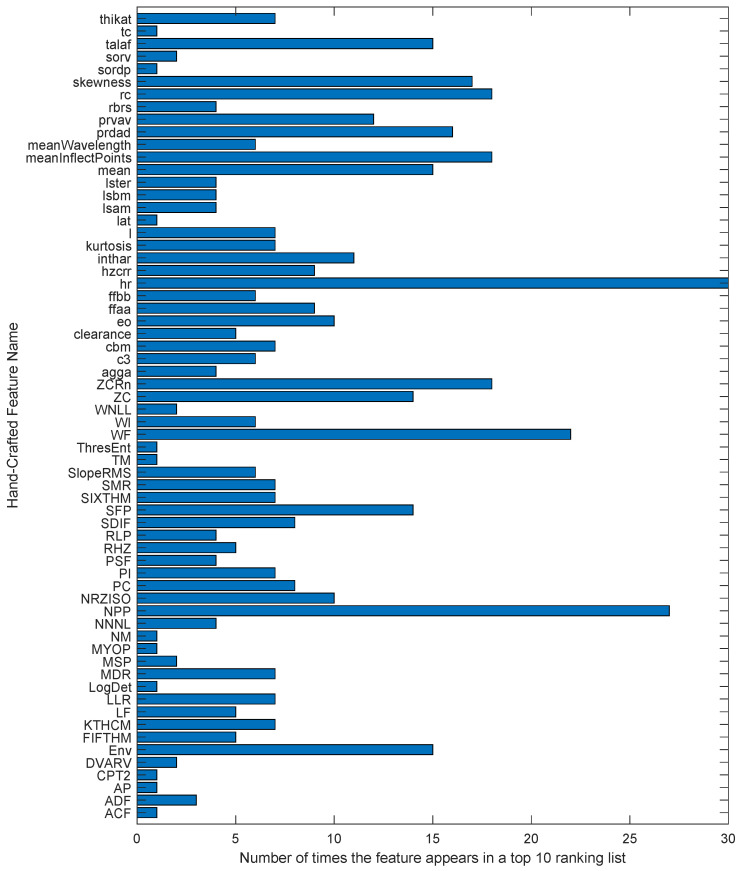
Hand-crafted feature occurrence distribution of the top ten ranking lists for all databases of the only time features group.

**Figure 8 sensors-24-05400-f008:**
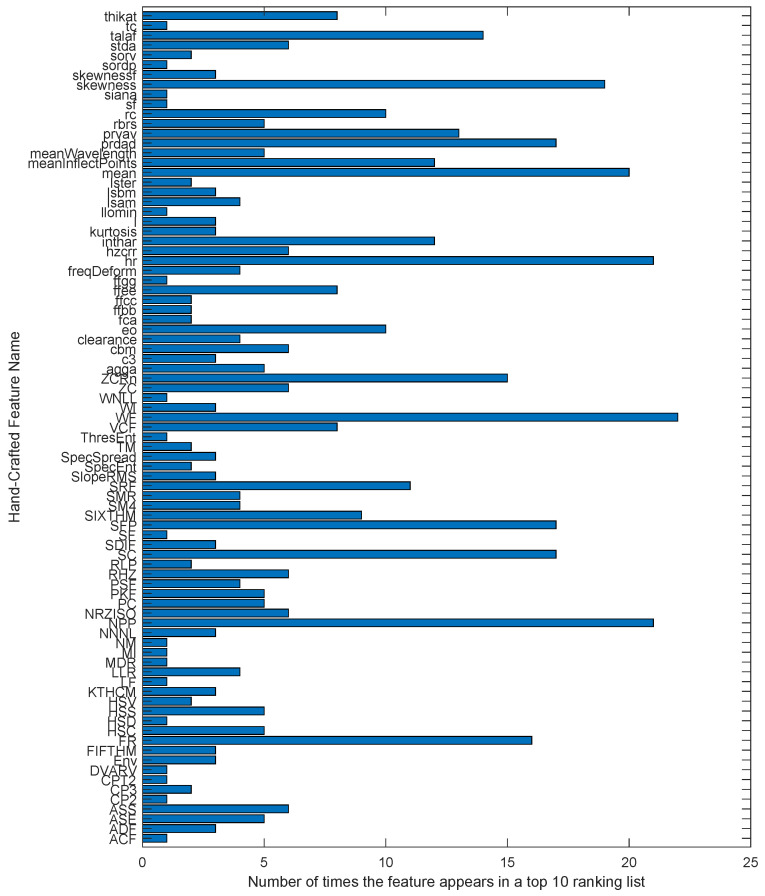
Hand-crafted feature occurrence distribution of the top ten ranking lists for all databases of the feature fusion group.

**Figure 9 sensors-24-05400-f009:**
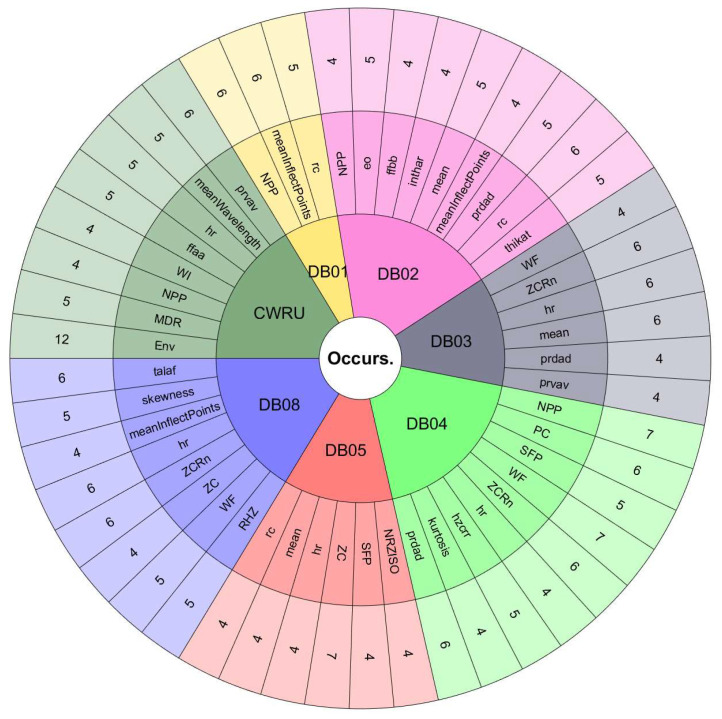
Occurrences in the top ten feature ranking lists of all databases for the group of only time features.

**Figure 10 sensors-24-05400-f010:**
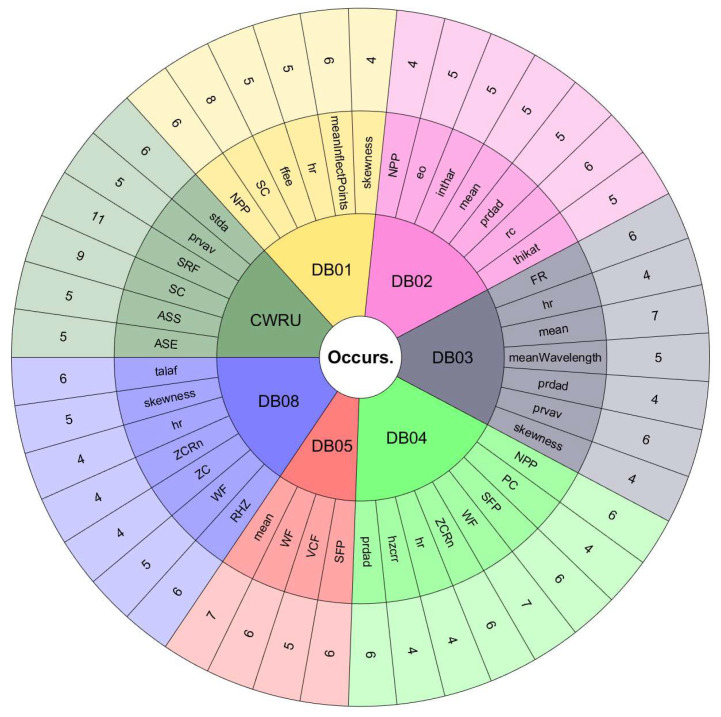
Occurrences in the top ten feature ranking lists of all databases for the group of fusion features.

**Figure 11 sensors-24-05400-f011:**
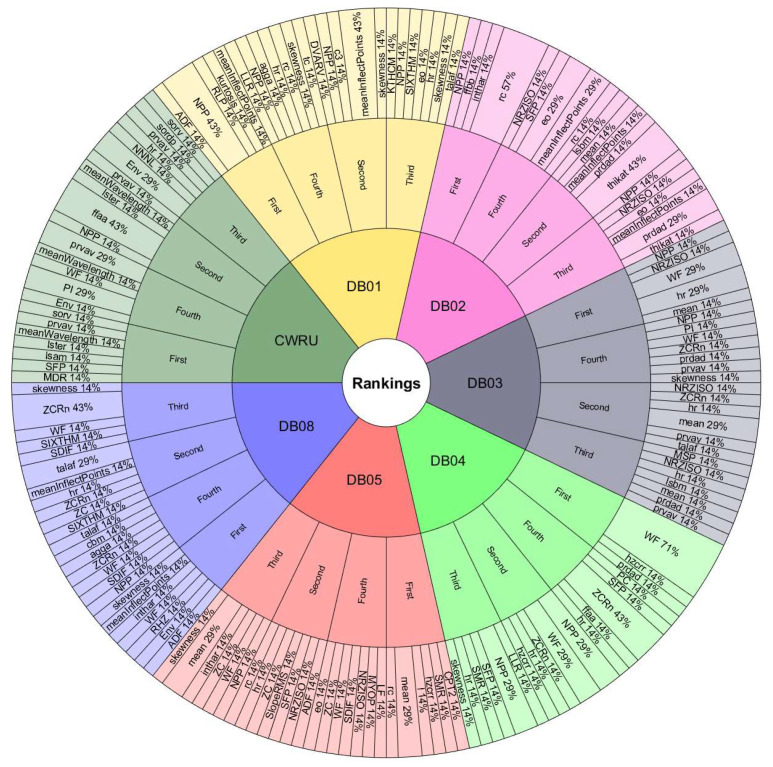
Sunburst plot of databases and feature apparition ratio in top four ranking importance in the only time features group. The second ring refers to the four most important positions in the rankings, and the third ring denotes the features placed there by the different ranking methods and the percentage of appearance in that position.

**Figure 12 sensors-24-05400-f012:**
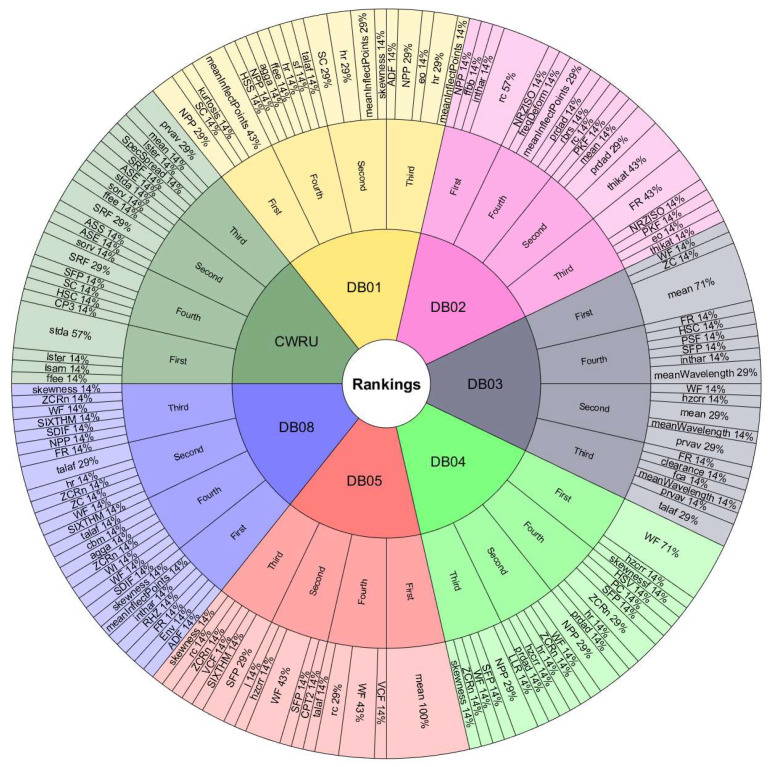
Sunburst plot of databases and feature apparition ratio in top four ranking importance in the fusion features group. The second ring refers to the four most important positions in the rankings, and the third ring denotes the features placed there by the different ranking methods and the percentage of appearance in that position.

**Figure 13 sensors-24-05400-f013:**
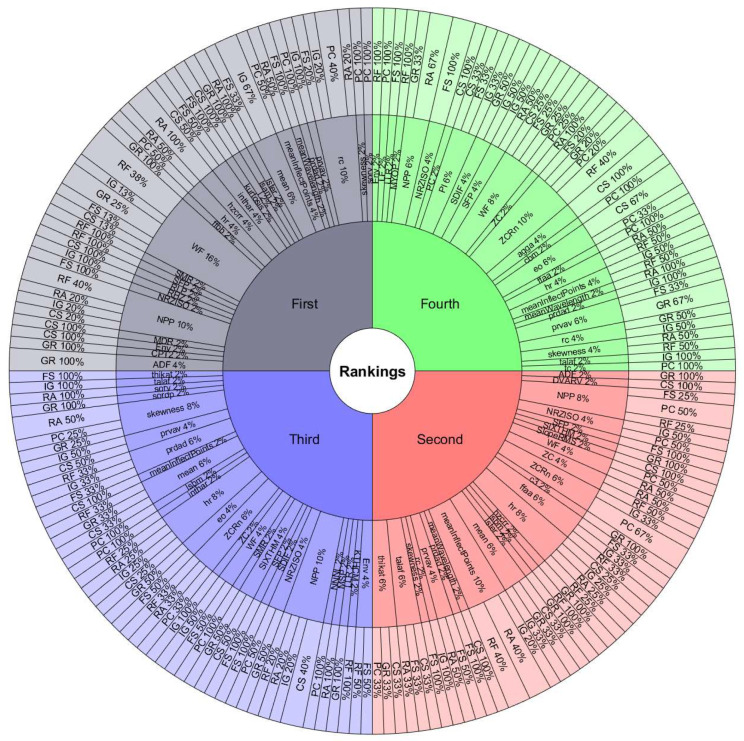
The top four ranked as the most important features and their appearance ratio in those places for all databases for the only time features group. The second ring denotes the percentage of occurrences of the feature in that position, while the third indicates the percentage of times the ranking method places the feature there.

**Figure 14 sensors-24-05400-f014:**
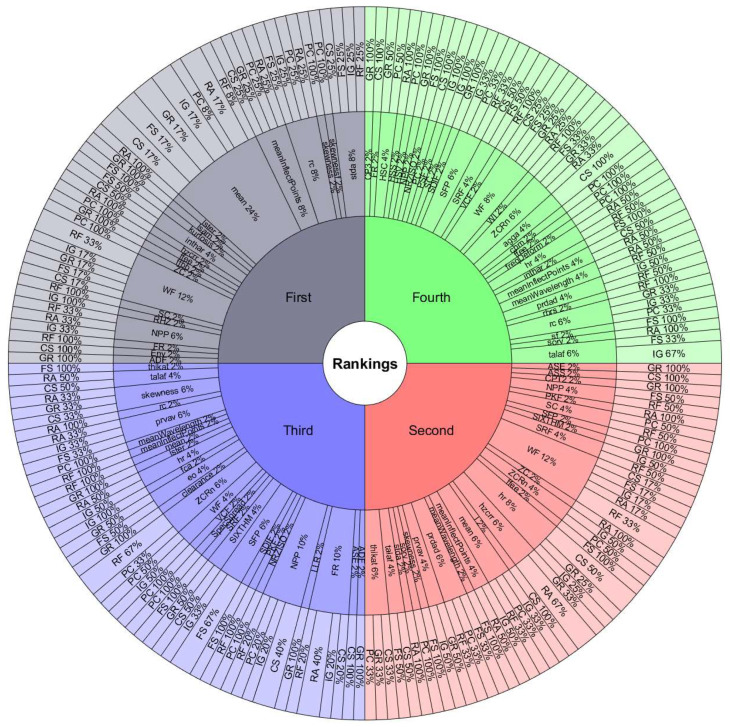
The top four ranked as the most important features and their appearance ratio in those places for all databases for the fusion features group. The second ring denotes the percentage of occurrences of the feature in that position, while the third indicates the percentage of times the ranking method places the feature there.

**Table 1 sensors-24-05400-t001:** Main review publication closer to hand-crafted feature extraction in rotating machinery.

Ref.	First Author	Year	Main Topic	Feature Extraction
[22]	H. Yang	2003	Rotating Machinery	TF: 7
[44]	P. Večeř	2005	Gearbox	TF: 6
[61]	W. Yan	2008	Bearings	TF: 5
[62]	K. Tom	2010	Bearings and Gears	TF: 8
[63]	A. S. Sait	2011	Gearbox	TF: 7
[64]	Z. Y. Han	2013	Gearbox	TF: 11
[65]	X. Zhao	2013	Planet gear teeth fault	TF: 18, FF: 30, PGF: 15
[10]	K. L. Tsui	2014	Data-driven approaches in Prognostics and Health Management	TF: 10
[66]	V. Sharma	2016	Gear condition indicators	TF: 14, FF: 7
[67]	W. Caesarendra	2017	Low-Speed Slew Bearing	TF: 9, FF: 6
[68]	S. Riaz	2017	Vibration Feature Extraction	TF: 7
[69]	A. Ogundare	2017	Helicopter Gearbox	TF: 3
[70]	D. Goyal	2017	Fixed Axis Gearbox	TF: 13, FF: 9, T-FF: 1
[71]	T. Wang	2019	Wind turbine planetary gearbox	TF: 14, FF: 4
[72]	A. Stetco	2019	Wind turbine	TF: 13
[73]	X. Zhang	2021	Bearings, Case Western Reserve University Data	TF: 5, FF: 5
[74]	M. A. Khan	2022	Bearings of Electrical Machines	TF: 7
[75]	S. Zhang	2022	Vibration signal processing in gears	TF: 5
[15]	S. Gawde	2023	Industrial Rotating Machines	TF: 13, FF: 6
[76]	R. Pandit	2023	SCADA data for wind turbines	TF: 3
[77]	M. Romanssini	2023	Vibration Monitoring of Rotating Machinery	TF:4
[78]	X. Xu	2024	Wind turbine gearbox	TF: 10, FF: 11
**This work**	**This work**	**2024**	**Rotating machinery**	**TF: 123, FF: 46**

**Table 2 sensors-24-05400-t002:** Feature nomenclature.

Symbol	Description
*y*	is a signal in the time domain
$y_{i}$	is the *i*-th element of *y*
*N*	is the total number of samples of *y*
*p*	is a threshold value
η	is a scale factor
$m_{t}$	is an integer value of the temporal moment order
$Rp_{i}$	is the peak average of the signal
$Rv_{i}$	is the valley average of the signal
$R3z_{i}T$	is the peak-to-peak value of the third ridge and valley
ROT	are the regions over a threshold
SBP	is the spacing between peaks
$N_{SBP}$	is the total number of spaces between peaks
$N_{HSC}$	is the total number of regions above threshold
g(x)	is a custom function
PPCM	is the spaces between profile peaks crossing the midline
lag	is a period between one event and another.
SAM	is a vector containing the lengths of consecutive signal samples above the mean.
SBM	is a vector containing the lengths of consecutive signal samples below the mean.
lb	is a lower bound
ub	is an upper-bound
*r*	is a multiplier of the standard deviation
*l*	is an integer number corresponding to the index of a time frame
$N_{hop}$	is the number of samples per analysis section
*L*	is *L* is the total number of time frames
$y_{l}$	is the *l*-th time frame from the *y* signal
$STE_{i}$	is the *i*-th of energy in a short period
avSTE	is the average STE over a 1 s window
$N_{w}$	is the number of points of a sample frame of the original signal $y_{i}$
$f_{0}$	is the minimum fundamental frequency to be analyzed
*Y*	is the frequency amplitude spectrum of *y*
$Y_{k}$	is the *k*-th measurement of the frequency amplitude spectrum (*Y*)
*K*	is the number of lines in the frequency spectrum
*P*	is the power spectrum frequency of *y*
$P_{k}$	is the *k*-th measurement of the power spectrum frequency (P)
$K_{P}$	is the total number of spectrum lines in the power spectrum
$f_{k}$	is the frequency value of the *k*-th spectrum line
ULC	Upper-cutoff frequency of the low-frequency band.
LLC	Lower-cutoff frequency of the low-frequency band.
UHC	Upper-cutoff frequency of the high-frequency band.
LHC	Lower-cutoff frequency of the high-frequency band.
$f_{h,l}$	is the frequency of harmonic peak *h*-th in frame *l*-th.
$N_{H}$	is the number of harmonics that is considered.
$A_{h,l}$	is the amplitude of harmonic *h*-th in frame *l*-th.
$S_{l}\left(k\right)$	is the *l*-th frame of a frequency spectrum
$N_{FT}$	is the number of points in the current frame of the spectrum
δ	is a small parameter to avoid calculation overflow.
*b*	is the band number
$loK_{b}$	is the integer frequency bin corresponding to the lower Edge of the band loFb.
$hiK_{b}$	is the integer frequency bin corresponding to the higher Edge of the band hiFb.
∗	is the multiplication symbol

**Table 3 sensors-24-05400-t003:** Summary of the Time Feature Set.

Feature Name	Formula
Mean	$T1=\frac{1}{N}\sum_{i=1}^{N}y_{i}$
Variance	$T2=\frac{1}{N}\sum_{i=1}^{N}{\left(y_{i}-T1\right)}^{2}$
Standard deviation (STD)	$T3=\sqrt{\frac{1}{N}\sum_{i=1}^{N}{\left(y_{i}-T1\right)}^{2}}$
Root mean square (RMS)	$T4=\sqrt{\frac{1}{N}\sum_{i=1}^{N}{\left(y_{i}\right)}^{2}}$
Max value	T5=max(y)
Kurtosis	$T6=\frac{N\sum_{i=1}^{N}{\left(y_{i}-T1\right)}^{4}}{{\left[\sum_{i=1}^{N}{\left(y_{i}-T1\right)}^{2}\right]}^{2}}$
Skewness	$T7=\frac{N\sum_{i=1}^{N}{\left(y_{i}-T1\right)}^{3}}{T3^{3}}$
Energy operator (EO)	$T8=\frac{N^{2}\sum_{i=1}^{N}{\left({\left(y_{i+1}\right)}^{2}-{\left(y_{i}\right)}^{2}-\mathrm{mean}\left({\left(y_{i+1}\right)}^{2}-{\left(y_{i}\right)}^{2}\right)\right)}^{4}}{{\left[\sum_{i=1}^{N}{\left({\left(y_{i+1}\right)}^{2}-{\left(y_{i}\right)}^{2}-\mathrm{mean}\left({\left(y_{i+1}\right)}^{2}-{\left(y_{i}\right)}^{2}\right)\right)}^{2}\right]}^{2}}$
Mean of absolute values (Mean abs.)	$T9=\frac{1}{N}\sum_{i=1}^{N}|y_{i}|$
Square root amplitude value (SRAV)	$T10={\left(\frac{\sum_{i=1}^{N}\sqrt{|y_{i}|}}{N}\right)}^{2}$
Shape factor (SF)	$T11=\frac{T4}{T9}$
Impulse factor (IF)	$T12=\frac{T5}{T9}$
Crest factor	$T13=\frac{T5}{T4}$
Clearance factor	$T14=\frac{T5}{\frac{1}{N}\sum_{i=1}^{N}{\left(y_{i}\right)}^{2}}$
CPT1	$T15=\frac{\sum_{i=1}^{N}\log(|y_{i}|+1)}{N\log(T3+1)}$
CPT2	$T16=\frac{\sum_{i=1}^{N}\exp(y_{i})}{N*\exp(T3)}$
CPT3	$T17=\frac{\sum_{i=1}^{N}\sqrt{|y_{i}|}}{N*T2}$
Mean Square Error (MSE)	$T18=\frac{1}{N}\sum_{i=1}^{N}{\left(y_{i}-T1\right)}^{2}$
Log-log ratio (LLR)	$T19=\frac{1}{\log(T3)}\sum_{i=1}^{N}\log(|y_{i}|+1)$
Standard Deviation Impulse Factor (SDIF)	$T20=\frac{T3}{T9}$
5th statistical Moment (FIFTHM)	$T21=\sum_{i=1}^{N}{\left(y_{i}-T1\right)}^{5}$
6th statistical Moment (SIXTHM)	$T22=\sum_{i=1}^{N}{\left(y_{i}-T1\right)}^{6}$
5th norm. moment (NM)	$T23=\frac{\frac{1}{N}\sum_{i=1}^{N}{\left(y_{i}-T1\right)}^{5}}{\sqrt{{\left(\frac{1}{N}\sum_{i=1}^{N}{\left(y_{i}-T1\right)}^{2}\right)}^{5}}}$
Kth central moment (KTHCM)	$T24=\mathrm{mean}\left[{\left(y_{i}-T1\right)}^{k}\right]$
	kissetto3
Pulse index (PI)	$T25=\frac{T5}{T1}$
Margin index (MI)	$T26=\frac{T5}{{\left(\frac{1}{N}\sum_{i=1}^{N}\sqrt{|y_{i}|}\right)}^{2}}$
Mean Deviation Ratio (MDR)	$T27=\frac{T1}{T3}$
Difference absolute variance value (DVARV)	$T28=\frac{1}{N-2}\sum_{i=1}^{N-1}{\left(y_{i+1}-y_{i}\right)}^{2}$
Min value	$T29=\min(y_{i})$
Peak Value	$T30=\frac{1}{2}\left[T5-T29\right]$
Peak to peak	T31=T5−T29
Hist. lower bound (Hist.LB)	$T32=T29-\frac{1}{2}\frac{T5-T29}{N-1}$
Hist. upper bound (Hist.UB)	$T33=T5+\frac{1}{2}\frac{T5-T29}{N-1}$
Latitude factor (LF)	$T34=\frac{\max(|y_{i}|)}{{\left(\frac{1}{N}\sum_{i=1}^{N}\sqrt{|y_{i}|}\right)}^{2}}$
Norm. N. Neg. Likelihood (NNNL)	$T35=\ln\left(\frac{T3}{T4}\right)$
Waveform indicators (WI)	$T36=\frac{T4}{T1}$
Shannon entropy	$T37=-\sum_{i=1}^{N}y_{i}^{2}*\log\left(y_{i}^{2}\right)$
Log energy entropy (LEE)	$T38=\sum_{i=1}^{N}\log(y_{i}^{2})$
	where log(0)=0
Threshold entropy	$T39=\left\{\begin{array}{cc} 1, & if\;|y_{i}|>p,\mathrm{and}\\ 0, & \mathrm{elsewhere} \end{array}\right.$
	*p* is set to 0.2
Sure entropy	T40=N−#{i such that
	$|y_{i}\left|\leq p\right\}+\ldots$$\ldots +\sum_{i}\min(y_{i}^{2},p^{2}),$ *p* is set to 0.2
Norm entropy	$T41=\sum_{i=1}^{N}|y_{i}{|}^{p},$
	*p* is set to 0.2
Slope sign change (SSC)	$T42=\sum_{i=2}^{N}g\left[\left(y_{i}-y_{i-1}\right)*\left(y_{i}-y_{i+1}\right)\right]$
	$g\left(y\right)=\left\{\begin{array}{c} 1,if\;\;\;y\geq \;p\\ 0,if\;\;\;\mathrm{otherwise} \end{array}\right.$
Zero crossing (ZC)	$T43=\sum_{i=1}^{N}\mathrm{step}[\mathrm{sign}\left(-y_{i}*y_{i+1}\right)]$
	$\mathrm{sign}=\left\{\begin{array}{c} 1,if\;\;y>0\\ 0,if\;\;y=0\\ -1,if\;\;y<0 \end{array}\right.$
	$\mathrm{step}=\left\{\begin{array}{cc} 1,if\;\;y>0\\ \frac{1}{2},if\;\;y=0 & \\ 0,if\;\;y<0 \end{array}\right.$
Wilson amplitude	$T44=\sum_{i=1}^{N}g(|y_{i}-y_{i+1}|-p)$
	$g\left(y\right)=\left\{\begin{array}{c} 1,if\;\;y\geq 0\\ 0,if\;y<0 \end{array}\right.$,
	*p* is set to 0.2
Myopulse percentage rate (MYOP)	$T45=\frac{1}{N}\sum_{i=1}^{N}\left[g\left(y_{i}\right)\right];$
	$g\left(y\right)=\left\{\begin{array}{c} \;1,\;\;\;if\;\;y\geq p\\ 0,\;if\;\;\mathrm{otherwise} \end{array}\right.$,
	*p* is set to 0.2
Wavelength	$T46=\sum_{i=1}^{N}|y_{i+1}-y_{i}|$
Log detector	$T47=\exp(\frac{1}{N})\sum_{i=1}^{N}\log|y_{i}|$
Mean of amplitude (MA)	$T48=\sum_{i=1}^{N}\left(y_{i-1}-y_{i}\right)$
Energy	$T49=\sum_{i=1}^{N}|y_{i}{|}^{2}$
Integrated signal	$T50=\sum_{i=1}^{N}|y_{i}|$
Modified mean absolute value 1	$T51=\frac{1}{N}\sum_{i=1}^{N}W_{i}\left|y_{i}\right|$
	$W_{i}=\left\{\begin{array}{c} \;1,\mathrm{if}\;0.25N\leq i\leq \;0.75N\\ 0.5,\;\mathrm{if}\;\;\mathrm{otherwise} \end{array}\right.$
Modified mean absolute value 2	$T52=\frac{1}{N}\sum_{i=1}^{N}W_{i}\left|y_{i}\right|$
	$W_{i}=\left\{\begin{array}{cc} \;\;\;1,\;\mathrm{if}\;0.25N\leq i\leq \;0.75N\\ \frac{4i}{N},\;\mathrm{if}\;\;i<0.25N & \\ \frac{4(i-N)}{N},\;\mathrm{if}\;\;\;i>0.75N \end{array}\right.$
Mean absolute value slope (MAVSLP)	$T53=T9_{i+1}-T9_{i}$
Delta RMS (DRMS)	$T54=T4_{i+1}-T4_{i}$
Root sum of squares (RSSQ)	$T55=\sqrt{\sum_{i=1}^{N}|y_{i}{|}^{2}}$
Weighted SSR absolute (WSSRA)	$T56=\frac{1}{N}{\left(\sum_{i=1}^{N}\sqrt{|y_{i}|}\right)}^{2}$
Log RMS	T57=log(T4)
Conduction velocity of Signal (CVS)	$T58=\frac{1}{N-1}\sum_{i=1}^{N}y_{i}^{2}$
Average amplitude change (AAC)	$T59=\frac{1}{N}\sum_{i=1}^{N-1}\left|y_{i+1}-y_{i}\right|$
Weibull negative log-likelihood (WNLL)	$T60=-\sum_{i=1}^{N}{\log[\left(T11*\eta \right)}^{-T1}*\ldots$
	$\ldots *|y_{i}{|}^{T1-1}\exp{\left(\frac{|y_{i}|}{\eta }\right)}^{T1}]$
V-ORDER 3	$T61=\sqrt[{3}]{\frac{1}{N}\sum_{i=1}^{N}y_{i}^{3}}$
Maximum Fractal Length (MFL)	$T62=\log_{10}\sqrt{\sum_{i=1}^{N-1}{\left(y_{i}-y_{i+1}\right)}^{2}}$
Difference Absolute STD (DASDV)	$T63=\sqrt{\frac{1}{N-1}\sum_{i=1}^{N-1}{\left(y_{i+1}-y_{i}\right)}^{2}}$
Higher order Temp. Moments (TM)	$T64=\left|\frac{1}{N}\sum_{i=1}^{N}y_{i}^{m_{t}}\right|$,
	$m_{t}$ is set to 3 as default.
Autocorrelation function (ACF)	$T65=\frac{1}{N-1}\sum_{i=1}^{N}y_{i}*y_{i-1}$
Amplitude density function (ADF)	$T66=\sqrt[{2}]{2\pi *T3}*\exp\left(\frac{T31}{2*T3^{2}}\right)$
High spot count (NROT)	$T67=\frac{1}{N}\sum \left(\mathrm{ROT}\right)$,
	the threshold can be set to 70% of the maximum value
Mean slope of the profile (SOP)	$T68=\frac{1}{N-1}\sum_{i=1}^{N}\left|y_{i+1}-y_{i}\right|$
Average wavelength (meanWavelenght)	$T69=\frac{2\pi *T1}{T68}$
Mean spacing of adjacent peaks (MSBP)	$T70=\frac{1}{N_{SBP}}\sum SBP$
Peaks mean values (NRZDIN)	$T71=\frac{1}{2N}\left(\sum_{i=1}^{N}Rp_{i}-Rv_{i}\right)$
Mean height of peaks (Rpm)	$T72=\frac{1}{N}\sum_{i=1}^{N}Rp_{i}$
Mean depth of valleys (Rvm)	$T73=\frac{1}{N}\sum_{i=1}^{N}Rv_{i}$
Third point rugosity mean (RHZ)	$T74=\frac{\sum_{i=1}^{N}R3z_{i}}{N}$
Number of peaks in profile (NPP)	T75=sumofpeaks
Mean spacing in the mean line (Sm)	T76=mean(PPCM)
Peak count (PC)	$T77=\frac{1}{T76}$
Profile solidity factor (PSF)	$T78=\frac{T29}{T5}$
Relative length of the profile (RLP)	$T79=\frac{1}{N}\sum_{i=1}^{N}\sqrt{{\left(y_{i+1}-y_{i}\right)}^{2}+1}$
Mean peak radius of curvature (Rp)	$T80=\frac{1}{N-2}\sum_{i=1}^{N-2}\frac{2y_{i}-y_{i-1}-y_{i+1}}{N^{2}}$
Stepness factor of the profile (SFP)	$T81=\frac{T1}{T76}$
RMS slope of the profile (SlopeRMS)	$T82=\sqrt[{2}]{\left(\frac{1}{N-1}\right)\sum_{i=1}^{N-1}{\left(y_{i}-y_{i-1}-\Theta m\right)}^{2}}$,
	where $\;\Theta m=\frac{1}{N-1}\sum_{i=1}^{N-1}y_{i}-y_{i-1}$
Mean spacing at mean line (SMR)	$T83=\frac{1}{N-1}\sum_{i=1}^{N-1}\tan^{-1}\left(y_{i+1}-y_{i}\right)$
Mean of inflection points	$T84=\frac{1}{N}\sum \left(\mathrm{total}\;\mathrm{of}\;\mathrm{inflection}\;\mathrm{points}\right)$
Height of irregularities (NRZISO)	$T85=\frac{1}{N}\left(\sum_{i=1}^{N}Rp_{i}-Rv_{i}\right)$
Waviness factor of the profile (WF)	$T86=\frac{1}{T1}\sum_{i=1}^{N-1}\sqrt[{2}]{{\left(y_{i+1}-y_{i}\right)}^{2}+1}$
Estimation of the Autocorrelation (AGGA)	T87= $\frac{1}{\left(N-lag\right){\left(T2\right)}^{2}}\sum_{i=1}^{N-lag}\left(y_{i}-T1\right)\left(y_{i+lag}-T1\right)$
C3	T88= $\frac{1}{\left(N-2lag\right)}\sum_{i=1}^{N-2lag}\left({\left(y_{i+2lag}\right)}^{2}\left(y_{i+lag}\right)\left(y_{i}\right)\right)$
Count above mean (CAM)	$T89=\sum_{i=1}^{N}\left(y_{i}>T1\right)$
Count below mean (CBM)	$T90=\sum_{i=1}^{N}\left(y_{i}<T1\right)$
First location of maximum (FLOM)	$T91=\frac{T5}{N}$
First location of minimum (FLOMIN)	$T92=\frac{T29}{N}$
Has duplicate (HD)	$T93=\sum_{i=1}^{N}\left(Vect_{unique}\left(y_{i}\right)∼=0\right)∼=N$
Has duplicate max (HDMAX)	T94= $\left\{\begin{array}{cc} 1\;or\;True, & if\;\left(T5=y_{1}\ldots y_{N}\right),“Twice”\\ 0\;or\;False, & else \end{array}\right.$
Has duplicate min (HDMIN)	T95= $\left\{\begin{array}{cc} 1\;or\;True, & if\;\left(T29=y_{1}\ldots y_{N}\right),“Twice”\\ 0\;or\;False, & else \end{array}\right.$
Large standard deviation (LSD)	T96= $\left\{\begin{array}{cc} 1\;or\;True, & if\;(T3>r(T5-T29))\\ 0\;or\;False, & else \end{array}\right.$
Last location of maximum (LLOM)	$T97=\frac{\mathrm{index}(T5)}{N}$
Last location of minimum (LLOMIN)	$T98=\frac{\mathrm{index}(T29)}{N}$
Longest strike above mean (LSAM)	T99=max(SAM)
Longest strike below mean (LSBM)	T100=max(SBM)
Mean second derivate central (MSDC)	$T101=\frac{1}{N}\sum_{i=1}^{N-1}\frac{1}{2}\left(\left(y_{i+2}\right)-2\left(y_{i+1}\right)+\left(y_{i}\right)\right)$
Percentage of recurring data points all datapoints (PRDAD).	$T102=\frac{\mathrm{length}(\mathrm{Repeated}\;\mathrm{values}\;\mathrm{in}\;\mathrm{signal})}{\mathrm{length}(\mathrm{signal})}$
Percentage of recurring values to all values (PRVAV).	$T103=\frac{\mathrm{length}(\mathrm{Repeated}\;\mathrm{values}\;\mathrm{in}\;\mathrm{signal})}{\mathrm{length}(Vect_{unique}\left(signal\right))}$
Range count (RC)	$T104=\sum (\left(y_{i}\geq lb\right)\left(y_{i}<ub\right))$
Ratio beyond r sigma (RBRS)	$T105=\frac{1}{N}\sum (|y_{i}-T1\left|\;>\;\left(r*T3\right)\right)$
Sum of recurring data points (SORDP)	T106= Sum of non-unique values
Sum of recurring values (SORV)	T107= Sum of the non-unique values count
Factor B (ffbb)	$T108=\frac{T6*T13}{T3}$
Talaf	T109=log(T6+T4)
Thikat	$T110=\log\left(T6^{T13}+T4^{T30}\right)$
Siana	$T111=\log\left(\frac{T13^{T6}}{T30^{T4}}\right)$
Inthar	$T112=\log\left(\frac{T30^{T6}}{T13^{T4}}*T12\right)$
Audio power (AP)	$T113_{l}=\frac{1}{N_{hop}}\sum_{i=1}^{N_{hop}-1}{\left|y_{\left(i+lN_{hop}\right)}\right|}^{2}$
Temporary bell (Env)	$T114_{l}=\sqrt{T114_{l}}$
Zero Crossing Rate for each Nhop (ZCRn)	T115= $\frac{1}{2N_{hop}}\sum_{i=1}^{N_{hop}}\mathrm{sign}\left(y_{Nhop_{i-1}}-y_{Nhop_{i-2}}\right)*F_{s}$
Simple quadratic integral (SSI)	$T116=\sum_{i=1}^{N}|y_{i}{|}^{2}$
Runtime log (LAT)	$T117=\log_{10}\left(\mathrm{index}\left(T5\right)-\mathrm{index}\left(T29\right)\right)$
Temporal centroid (TC)	$T118=\frac{N_{hop}}{F_{s}}\frac{\sum_{l=1}^{L}\left(lT115_{l}\right)}{\sum_{l=1}^{L}\left(T115_{l}\right)}$
Harmonic Ratio (HR)	$\Gamma_{m}=\frac{\sum_{i=1}^{N_{w}}\left(y_{i}\right)*\left(y_{i-m}\right)}{\sqrt{\sum_{i=1}^{N_{w}}{\left(y_{i}\right)}^{2}*\sum_{i=1}^{N_{w}}{\left(y_{i-m}\right)}^{2}}}$,
	(1≤m≤M;0≤l≤(L−1)),
	where $M=\frac{F_{s}}{{\left(f_{0}\right)}^{2}}$, is set to 2 lags
	$T119=max\left(\Gamma_{m}\right)\;\left(1\leq m\leq M\right)$
High Zero-Crossing Rate Ratio (HZCRR)	$T120=\frac{1}{2L}\sum_{i=1}^{L}[\mathrm{sign}(T116+\ldots$
	…−1.5∗mean(T116))+1]
Low energy ratio in the short term (LSTER)	T121= $\frac{1}{2L}\sum_{i=1}^{L}\left[\mathrm{sign}(0.5*avSTE-STE_{i})+1\right]$
Health indicators (INDI)	$T122=\frac{T4}{T3}$
Factor A (ffaa)	$T123=\frac{T5}{T3*T2^{2}}$

**Table 5 sensors-24-05400-t005:** Summary of the Frequency Feature Set.

Feature Name	Formula
Mean Frequency	$F1=\frac{\sum_{k=1}^{K}Y_{k}}{K}$
Variancef	$F2=\frac{\sum_{k=1}^{K}{\left(Y_{k}-F1\right)}^{2}}{K-1}$
Skewnessf	$F3=\frac{\sum_{k=1}^{K}{\left(Y_{k}-F1\right)}^{3}}{K{\left(\sqrt{F2}\right)}^{3}}$
Kurtosisf	$F4=\frac{\sum_{k=1}^{K}{\left(Y_{k}-F1\right)}^{4}}{K{\left(F2\right)}^{4}}$
Central Frequency	$F5=\frac{\sum_{k=1}^{K}f_{k}Y_{k}}{\sum_{k=1}^{K}Y_{k}}$
STDF	$F6=\sqrt{\frac{\sum_{k=1}^{K}{\left(f_{k}-F5\right)}^{2}Y_{k}}{\sum_{k=1}^{K}Y_{k}}}$
RMSF	$F7=\sqrt{\frac{\sum_{k=1}^{K}f_{k}^{2}Y_{k}}{\sum_{k=1}^{K}Y_{k}}}$
CP1	$F8=\frac{\sum_{k=1}^{K}{\left(f_{k}-F5\right)}^{3}Y_{k}}{K{\left(F6\right)}^{3}}$
CP2	$F9=\frac{F6}{F5}$
CP3	$F10=\frac{\sum_{k=1}^{K}{\left(f_{k}-F5\right)}^{\frac{1}{2}}Y_{k}}{K\sqrt{F6}}$
CP4	$F11=\frac{\sum_{k=1}^{K}{\left(f_{k}-F5\right)}^{3}Y_{k}}{F6^{2}K}$
CP5	$F12=\sqrt{\frac{\sum_{k=1}^{K}f_{k}^{4}Y_{k}}{\sum_{k=1}^{K}f_{k}^{2}Y_{k}}}$
Spectral Centroid	$F13=\frac{\sum_{k=1}^{K}kY_{k}}{\sum_{k=1}^{K}Y_{k}}$
Spectral Spread	$F14=\sqrt{\frac{\sum_{k=1}^{K}{\left(k-F13\right)}^{2}Y_{k}}{\sum_{k=1}^{K}Y_{k}}}$
Spectral Entropy	$F15=-\sum_{k=1}^{K-1}P_{n}\left(k\right)\log_{2}\left[P_{n}\left(k\right)\right]$
	where:
	$P_{n}\left(k\right)=\frac{Y_{k}}{\sum_{k=1}^{K}Y_{k}}$
Total power	$F16=\sum_{k=1}^{K_{P}}P_{k}$
Median Frequency	$F17=\frac{1}{2}\sum_{k=1}^{K_{P}}P_{k}$
Peak frecuency (PKF)	F18=max(P)
First Spectral Moment	$F19=\sum_{k=1}^{K_{P}}P_{k}f_{k}$
Second Spectral Moment	$F20=\sum_{k=1}^{K_{P}}P_{k}f_{k}^{2}$
Third Spectral Moment	$F21=\sum_{k=1}^{K_{P}}P_{k}f_{k}^{3}$
Fourth Spectral Moment	$F22=\sum_{k=1}^{K_{P}}P_{k}f_{k}^{4}$
Variance of central frequency(VCF)	$F23=\frac{F20}{F16}-{\left(\frac{F19}{F16}\right)}^{2}$
Frequency Deformation	$F24=\frac{\sqrt{F20/F16}}{F19/F16}$
Frequency ratio (FR)	$F25=\sum_{LLC=f_{min}}^{ULC=f_{max}/2}P_{k}/\sum_{LHC=\frac{f_{max}}{2}+1}^{UHC=f_{max}}P_{k}$
Harmonic Spectral Centroid (HSC)	$F26=\frac{1}{L}\sum_{l=1}^{L}LHSC_{l}$,
	where $LHSC_{l}=\frac{\sum_{h=1}^{N_{H}}\left(f_{h,l}A_{h,l}\right)}{\sum_{h=1}^{N_{H}}\left(A_{h,l}\right)}$
Harmonic Spectral Deviation (HSD)	$F27=\frac{1}{L}\sum_{l=1}^{L}LHSD_{l}$,
	where $LHSD_{l}=\frac{\sum_{h=1}^{N_{H}}\left|\log_{10}\left(A_{h,l}\right)-\log_{10}\left(SE_{h,l}\right)\right|}{\sum_{h=1}^{N_{H}}\log_{10}\left(A_{h,l}\right)}$,
	$SE_{h,l}=\left\{\begin{array}{cc} \frac{1}{2}\left(A_{h,l}+A_{h+1,l}\right), & if\;h=1\\ \frac{1}{3}\left(A_{h-1,l}+A_{h,l}+A_{h+1,l}\right), & if\;2\leq h\leq N_{H}-1\\ \frac{1}{2}\left(A_{h-1,l}+A_{h,l}\right), & if\;h=N_{H} \end{array}\right.$
Harmonic Spectral Spread (HSS)	$F28=\frac{1}{L}\sum_{l=1}^{L}LHSS_{l}$,
	where
	$LHSS_{l}=\frac{1}{LHSC_{l}}\sqrt{\frac{\sum_{h=1}^{N_{H}}\left[{\left(f_{h,l}-LHSC_{l}\right)}^{2}A_{h,l}^{2}\right]}{\sum_{h=1}^{N_{H}}A_{h,l}^{2}}}$
Harmonic Spectral Variation (HSV)	$F29=\frac{1}{L}\sum_{l=1}^{L}LHSV_{l}$,
	where
	$LHSV_{l}=1-\frac{\sum_{h=1}^{N_{H}}\left(A_{h,l-1}A_{h,l}\right)}{\sqrt{\sum_{h=1}^{N_{H}}A_{h,l-1}^{2}}\sqrt{\sum_{h=1}^{N_{H}}A_{h,l}^{2}}}$
Spectral Flux (SF)	F30= $\frac{1}{L*N_{FT}}\sum_{k=1}^{L}\sum_{k=1}^{N_{FT}}\left[\log(\right|S_{l}\left(k\right)|+\delta )-\log(|S_{l-1}{\left(k\right)|+\delta )]}^{2}$
Frequency Centre (FC)	$F31=\frac{\sum_{k=2}^{K}x_{k^{\prime }}x_{k}}{2\pi \sum_{k=1}^{K}x_{k}^{2}}$,
	where
	$x_{k^{\prime }}=x\left(k+1\right)-x_{k}$
Mean square Frequency (MSF)	$F32=\frac{\sum_{k=2}^{K}{\left(x_{k^{\prime }}\right)}^{2}}{4\pi^{2}\sum_{k=1}^{K}x_{k}^{2}}$
Root Mean square Frequency (RMSF)	$F33=\sqrt{F32}$
Grand mean (GM)	$F34=\frac{\sum_{k=1}^{K}f_{k}Y_{k}}{\sum_{k=1}^{K}Y_{k}}$
Standard Deviation (STDA)	$F35=\sqrt{\frac{\sum_{k=1}^{K}{\left(f_{k}-F35\right)}^{2}Y_{k}}{K}}$
C Factor (ffcc)	$F36=\sqrt{\frac{\sum_{k=1}^{K}f_{k}^{2}Y_{k}}{\sum_{k=1}^{K}Y_{k}}}$
D Factor (ffdd)	$F37=\sqrt{\frac{\sum_{k=1}^{K}f_{k}^{4}Y_{k}}{\sum_{k=1}^{K}f_{k}^{2}Y_{k}}}$
E Factor (ffee)	$F38=\frac{\sum_{k=1}^{K}f_{k}^{2}Y_{k}}{\sqrt{\sum_{k=1}^{K}Y_{k}\sum_{k=1}^{K}f_{k}^{4}Y_{k}}}$
G Factor (ffgg)	$F39=\frac{F36}{F35}$
Audio Spectrum Envelope (ASE)	$F40=\sum_{k=1}^{N_{FT}}P_{b}\left(k\right)$, for 1≤b≤B
Audio Spectrum Flatness (ASF)	$F41=\frac{\sqrt[{hiK_{b}^{\prime }-loK_{b}^{\prime }+1}]{\prod_{k^{\prime }=loK_{b}^{\prime }}^{hiK_{b}^{\prime }}}P_{b}\left(k\right)}{\frac{1}{hiK_{b}^{\prime }-loK_{b}^{\prime }+1}\sum_{k^{\prime }=loK_{b}^{\prime }}^{hiK_{b}^{\prime }}P_{b}\left(k\right)}$, for 1≤b≤B
Audio Spectrum Spread (ASS)	$F42=\sqrt{\frac{\sum_{k^{\prime }=0}^{\left(N_{FT}/2\right)-K_{low}}{\left[\log_{2}\left(\frac{f^{\prime }\left(k^{\prime }\right)}{1000}\right)-F42\right]}^{2}P^{\prime }\left(k^{\prime }\right)}{\sum_{k^{\prime }=0}^{\left(N_{FT}/2\right)-K_{low}}P^{\prime }\left(k^{\prime }\right)}}$
Power Spectral Centroid Segment (SC)	$F43=\frac{\sum_{k=0}^{N_{FT}}f\left(k\right)P_{s}\left(k\right)}{\sum_{k=0}^{N_{FT}}P_{s}\left(k\right)}$, where
	$P_{s}$ is the estimated power spectrum for the segment.
SNR	$F44=\frac{F1}{F36}$
Spectral Rolloff Frequency (SRF)	$F45=0.85\sum_{k=0}^{N_{FT}}\left|Y_{k}\right|$
Upper Limit of Harmonicity (ULH)	$F46=\log_{2}\left(\frac{f_{ulh}}{1000}\right)$,
	where $f_{ulh}$ is defined by:
	$f_{ulh}=\left\{\begin{array}{cc} 31.25, & for\;k^{\prime }=0\\ f(k^{\prime }+K_{low}), & for\;1<k^{\prime }<\frac{N_{FT}}{2}-K_{low} \end{array}\right.$

**Table 6 sensors-24-05400-t006:** Summary of classification results of all experimentation for spur gear gearboxes.

Database	Ranking Method	Only Time Features						Only Frequency Features						Fusion Time and Frequency Features					
		RF		KNN		SVM		RF		KNN		SVM		RF		KNN		SVM	
		Cl. %	#F.	Cl. %	#F.	Cl. %	#F.	Cl. %	#F.	Cl. %	#F.	Cl. %	#F.	Cl. %	#F.	Cl. %	#F.	Cl. %	#F.
**DB01**	RA	86.61	10	86.24	10	38.91	10	68.19	10	67.43	10	56.38	10	82.89	10	82.69	10	49.33	10
	CS	84.7	9	81.93	10	42.06	9	68.95	10	62.19	10	55.52	9	84.09	10	80.55	7	46.66	10
	IG	82.31	6	82.98	10	36.43	10	68.57	10	68.95	10	59.81	10	81.88	10	79.75	9	53	10
	PC	71.89	8	70.94	6	35.76	9	50.1	9	38.67	4	33.43	7	72.06	10	71.19	10	45.12	9
	FS	69.5	7	57.27	7	23.89	8	69.52	10	62.57	7	30.57	10	64.03	8	49.94	8	16.31	10
	GR	82.22	8	70.75	10	28.49	9	64.48	10	60.38	10	54.1	8	82.55	9	74.4	10	39.84	8
	RF	87.09	8	79.73	8	36.15	9	69.33	10	69.71	10	59.52	9	82.09	9	80.95	10	56.01	10
**DB03**	RA	97.66	9	96.88	7	72.94	10	94.11	8	95.33	5	60.89	9	64.59	9	57.61	8	56.08	8
	CS	94.99	8	89.09	10	67.15	10	93.78	10	86	10	48.11	10	68.85	8	55.8	7	56.18	8
	IG	97.33	10	98.77	10	78.51	10	94.56	8	96.56	9	62	10	69.89	10	44.45	1	46.57	10
	PC	74.83	8	65.03	8	33.41	10	49.67	7	42	9	35.11	10	58.91	6	45.41	2	53.96	10
	FS	94.65	8	93.21	8	78.06	10	93.89	7	86.89	3	52.22	10	65.76	10	51.61	9	50.24	10
	GR	93.21	10	90.43	10	71.05	10	90.56	8	86.56	6	42.11	9	60.13	10	45.62	2	50.03	10
	RF	94.32	9	93.43	10	70.38	10	90.78	10	91.56	4	61.11	8	68.04	9	56.62	10	54.11	10

**Table 7 sensors-24-05400-t007:** Summary of classification results of all experimentation for helical gear gearboxes.

Database	Ranking Method	Only Time Features						Only Frequency Features						Fusion Time and Frequency Features					
		RF		KNN		SVM		RF		KNN		SVM		RF		KNN		SVM	
		Cl. %	#F.	Cl. %	#F.	Cl. %	#F.	Cl. %	#F.	Cl. %	#F.	Cl. %	#F.	Cl. %	#F.	Cl. %	#F.	Cl. %	#F.
**DB04**	RA	96.16	7	97.58	10	75.76	10	85.86	5	89.9	4	55.56	10	96.97	10	97.37	9	78.99	10
	CS	97.17	9	97.17	10	75.35	10	82.02	8	73.54	8	53.94	9	98.18	9	98.59	9	72.53	10
	IG	96.57	8	97.37	9	78.38	9	84.04	7	77.17	7	53.13	10	97.17	9	97.37	10	80	10
	PC	97.37	9	97.78	7	71.92	9	71.92	8	45.66	8	34.34	10	93.13	10	82.63	10	64.85	10
	FS	96.57	9	96.36	8	75.56	9	76.36	9	64.44	4	46.87	9	96.77	9	96.77	9	75.76	10
	GR	97.58	9	95.76	9	73.33	10	68.69	5	55.76	5	42.63	9	97.17	10	95.96	9	73.33	10
	RF	97.98	9	98.59	8	78.59	10	83.43	9	75.35	7	55.56	10	96.97	9	97.58	9	86.26	10
**DB05**	RA	93.07	10	87.2	10	41.07	10	70	7	66	7	23.47	9	86.53	10	87.47	10	44	10
	CS	90	10	81.73	10	38.27	10	61.87	9	52.27	9	23.6	10	89.47	10	84.93	9	52.13	10
	IG	86.13	9	79.07	9	44.53	10	72	7	65.87	7	20.8	10	86.93	10	84.4	10	55.33	10
	PC	90.13	9	83.33	10	36.8	10	44.13	6	34.4	3	18.93	9	87.47	9	87.2	9	50.93	10
	FS	86.67	10	77.07	9	42.93	10	46.53	10	40.67	10	18.93	9	84.53	10	80.53	8	54.4	10
	GR	77.87	9	66.27	10	34.93	10	32.13	6	30.8	6	17.73	10	85.73	10	62.53	8	28.93	8
	RF	91.87	7	93.07	7	37.2	8	71.87	8	65.47	8	21.47	8	84.13	10	86	10	43.33	9
**DB08**	RA	98	9	98	10	60.76	10	90.33	9	90.56	5	39.67	10	98.44	10	98.78	10	64.22	10
	CS	92.78	10	94.27	10	36.52	10	91.44	7	91.78	6	36	8	93	10	93.72	10	36.83	10
	IG	97.67	10	98.61	10	54.81	10	90.56	8	90.72	5	39.06	10	98	10	98.39	9	58.33	10
	PC	96	10	95.89	10	45.41	10	57.72	10	35.39	10	20.61	10	94.39	10	91.5	10	38.56	10
	FS	98	10	98.22	10	55.2	10	83.56	8	76	7	36.28	9	97.72	9	98.61	10	57.11	10
	GR	97.33	9	97.83	10	49.64	10	90.11	10	84.56	6	37.56	10	97.17	9	97.67	10	50.06	10
	RF	98.39	9	99	10	61.98	10	90.67	9	90.61	5	39.56	10	98	10	99	10	67.67	10

**Table 8 sensors-24-05400-t008:** Summary of classification results of all experimentation for rotating machinery bearings.

Database	Ranking Method	Only Time Features						Only Frequency Features						Fusion Time and Frequency Features					
		RF		KNN		SVM		RF		KNN		SVM		RF		KNN		SVM	
		Cl. %	#F.	Cl. %	#F.	Cl. %	#F.	Cl. %	#F.	Cl. %	#F.	Cl. %	#F.	Cl. %	#F.	Cl. %	#F.	Cl. %	#F.
**CWRU**	RA	99.37	6	99.05	7	98.75	8	99.69	7	100	10	99.69	5	99.36	2	99.68	8	99.68	6
	CS	98.74	10	99.05	3	97.81	9	99.69	8	99.38	3	99.69	8	99.69	7	99.06	8	100	9
	IG	99.68	8	98.73	9	99.06	8	99.69	7	100	9	99.69	10	100	9	100	10	99.38	9
	PC	98.1	4	98.11	6	98.44	7	100	7	100	10	100	9	100	8	100	5	100	10
	FS	100	5	99.36	9	99.06	6	100	5	100	7	100	4	99.68	8	100	6	100	4
	GR	99.68	7	99.05	7	99.06	8	99.69	4	99.38	5	99.69	6	100	9	99.05	8	99.69	10
	RF	99.69	8	99.06	4	98.75	7	99.69	9	99.38	10	100	9	100	3	100	2	100	6
**DB02**	RA	89.48	9	87.24	10	66.77	10	84.76	6	80.95	4	40	9	92.66	10	91.37	9	67.74	10
	CS	87.54	9	89.12	7	60.38	9	84.76	5	85.4	5	41.27	10	90.43	10	88.19	7	58.78	10
	IG	89.8	8	87.23	10	51.11	10	85.08	5	83.17	5	40	5	92.03	10	92.35	9	60.39	10
	PC	76.06	8	64.25	8	54.63	8	82.22	4	79.05	4	37.14	10	78.29	10	65.49	10	58.44	9
	FS	75.4	7	67.43	7	52.67	10	83.81	6	83.49	4	38.41	9	77.34	9	70.6	8	52.69	7
	GR	79.89	10	73.78	10	49.17	9	82.22	4	83.17	4	40.63	9	82.14	10	71.53	10	47.93	9
	RF	92.05	9	90.71	9	67.73	10	84.76	5	86.03	5	40	7	86.91	9	91.04	10	65.17	10

**Table 9 sensors-24-05400-t009:** Summary of the best results settings for each database.

Mech. Comp.	Spur Gear Gearboxes				Helical Gear Gearboxes						Bearings			
**Database**	DB01		DB03		DB04		DB05		DB08		DB02		CWRU	
**Eval. group**	Only time		Only time		Only time		Only time		Fusion		Fusion		Fusion	
**Max class %**	87.09		98.77		98.59		93.07		99.00		92.66		100.00	
**Acc. std**	1.61		0.25		1.15		2.14		0.54		2.67		0.00	
**Class Model**	RF		KNN		KNN		RF		KNN		RF		KNN	
**Ranking Method**	RF		IG		RF		RA		RF		RA		RF	
**1st feat.**	NPP	24.18	mean	39.31	WF	45.25	hzcrr	14.67	FR	20.39	rc	28.78	stda	99.05
**2nd feat.**	meanIn- flectPoints	36.42	NRZISO	69.26	NPP	69.49	hr	31.07	WF	47.39	PKF	58.17	**SRF**	**100.00**
**3rd feat.**	KTHCM	65.68	prvav	78.39	hr	87.27	skewness	63.73	NPP	78.33	FR	73.78	mean	99.37
**4th feat.**	hr	72.85	prdad	86.41	ZCRn	93.54	ZC	76.40	WI	92.28	meanIn- flectPoints	85.63	SFP	98.73
**5th feat.**	RLP	81.83	hr	94.10	SMR	95.96	rc	85.33	talaf	95.56	SFP	88.50	WF	98.75
**6th feat.**	eo	83.17	Env_3	95.10	prdad	98.38	NRZISO	89.20	SM4	96.56	NPP	89.79	HSS	99.05
**7th feat.**	c3	84.22	kurtosis	96.33	PC	96.77	PSF	90.13	PKF	97.67	mean	90.43	ASE	98.74
**8th feat.**	**DVARV**	**87.09**	Env_4	96.44	**kurtosis**	**98.59**	cbm	88.53	SlopeRMS	97.56	lsbm	90.44	Env_5	99.37
**9th feat.**	SMR	86.52	MSP	98.22	hzcrr	97.58	talaf	91.33	skewness	98.83	prdad	91.39	ASS	99.69
**10th feat.**	ffaa	85.56	**meanIn-** **flectPoints**	**98.77**	SFP	97.37	**NPP**	**93.07**	**RHZ**	**99.00**	**lsam**	**92.66**	prvav	99.68

Results highlighted in bold indicate the highest classification percentage achieved up to the tenth feature.

**Table 10 sensors-24-05400-t010:** Summary of the worst results settings for each database.

Mech. Comp.	Spur Gear Gearboxes				Helical Gear Gearboxes						Bearings			
**Database**	DB01		DB03		DB04		DB05		DB08		DB02		CWRU	
**Best Group**	Only time		Only time		Only Freq.		Only Freq.		Only Freq.		Only Freq.		Only time	
**Max class %**	23.89		33.41		34.34		17.73		20.61		37.14		97.81	
**Acc std**	4.87		3.55		5.76		1.12		2.23		6.79		4.89	
**Class Model**	SVM		SVM		SVM		SVM		SVM		SVM		SVM	
**Ranking Method**	FS		PC		PC		GR		PC		PC		CS	
**1st feat.**	kurtosis	14.72	hr	18.82	skewnessf	15.96	HSC_5	10.13	HSS_5	11.89	freqDeform	14.60	MDR	71.21
**2nd feat.**	skewness	15.11	ZCRn_3	25.05	HSV_1	15.15	HSV_4	10.40	HSC_4	10.67	SM4	18.10	meanWa- velength	92.74
**3rd feat.**	eo	16.73	lsbm	25.73	ASS_1	15.15	VCF	11.47	SC_4	10.89	PKF	26.98	prvav	91.75
**4th feat.**	LLR	17.89	ZCRn_2	26.39	HSD_1	16.57	HSS_4	11.20	SF_2	11.39	VCF	33.33	PI	90.82
**5th feat.**	SDIF	22.18	ZCRn_4	28.18	freqDeform	20.81	HSC_3	13.73	freqDeform	13.00	CP4	31.11	Env_2	97.18
**6th feat.**	FIFTHM	20.65	lsam	27.73	rmsfk	23.23	PKF	12.27	HSS_1	14.06	SF_3	32.38	SIXTHM	97.17
**7th feat.**	SIXTHM	20.37	ZCRn_1	29.62	PKF	25.66	SF_3	13.73	SF_3	17.28	SF_4	31.43	Env_4	97.17
**8th feat.**	**NM**	**23.89**	ZCRn_5	27.95	VCF	33.94	ffdd	11.73	HSV_2	16.94	SF_5	36.51	ffaa	97.17
**9th feat.**	MDR	19.69	lster	32.29	ASS_2	33.33	SF_5	14.93	SM4	19.72	SF_2	34.92	**inthar**	**97.81**
**10th feat.**	KTHCM	21.22	**lat**	**33.41**	**ASS_5**	**34.34**	**HSC_4**	**17.73**	**ffgg**	**20.61**	**HSS_1**	**37.14**	hr	97.81

Results highlighted in bold indicate the highest classification percentage achieved up to the tenth feature.

## Data Availability

The data presented in this study are available on request from the corresponding author due to privacy policy.

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
