# Peer review of "Evaluation of Hand-Crafted Feature Extraction for Fault Diagnosis in Rotating Machinery: A Survey"

_sensors, 2024, doi:10.3390/s24165400_

Round 1

Reviewer 1 Report

Comments and Suggestions for Authors

In the introduction section, the research objectives and contributions of this paper should be stated, the reasons why manual feature extraction is still of research value should be explained, and the main contents of this paper should be summarized.

In this paper, we use hand-calculated signal features, so how to deal with the accompanying noise for real working conditions?

In the feature extraction stage in the third part of the paper, data fusion is used to fuse the time domain features with the frequency domain features, please explain how it is done?

For each manual feature, explain how these features reflect the characteristics of the vibration signal

Describe the ranking methods and classifiers used in the feature selection process, providing the theoretical basis and reasons for the selection of each method.

In the experimental part it is recommended to compare the results with those of automated feature extraction methods and to analyze the differences between the two, instead of just comparing manual features.

In the Results and Discussion section, the experimental results are analyzed in depth to explain how each feature performs in different diagnostic tasks of faults and to find out which features are the most representative.

In the conclusion section, the main findings of this paper should be summarized, the advantages of manual feature extraction should be repeated, and future research directions can be briefly stated, for example, they can start from combining with automated feature extraction.

Reviewer 2 Report

Comments and Suggestions for Authors

This paper provides a comprehensive set of formulas and calculations for manually extracting the features of state monitoring signals. The paper is described in detail and has sufficient experiments.

1The conclusion part is too cumbersome. It is recommended to refine and simplify it;

2Please check the formulas in Table 3 and Table 4 to ensure the correctness of the formulas;

3Is it reasonable to only use three machine classification models for evaluation? It is recommended to increase machine classification models, especially deep learning models.

Reviewer 3 Report

Comments and Suggestions for Authors

The present work presents a comprehensive collection of "hand-crafted" features utilized in the condition monitoring of rotating machinery. The authors justify the novelty of the research by comparing the collection with other published surveys. The work's content is of interest since it presents a vast amount of useful content. The analyzed features are tested in several datasets of mechanical defects, including the commonly utilized baseline dataset CWRU.

Nevertheless, some minor modifications should be performed prior to publication:

- The article should emphasize in the title the "survey" or "review" nature of the contribution. This is a critical point to provide readers with the specific scope of the manuscript

- Contribution 2 should be reconsidered since the classification methodology is not a novel contribution of the manuscript.

- Some redundant information was found over the manuscript reading. Please, carefully review the redundancy of the information.

- Table 2 presents a glossary. This is normally placed in a dedicated section in the end of the manuscript.

- The classification algorithm steps from figure 5 do not coincide with the bullet points in the main text body. How are the data-fusion features extracted? Further describe this methodology.

- Tables 6 to 9. The text within the tables looks extremely small sized.

Comments on the Quality of English Language

Some typos and incorrect structures were found. Carefully review the manuscript.
